# Volatility Parameterization of Ambient Organic Aerosols at a rural site of the Northern China Plain

**Siman Ren[1], Lei Yao[1], Yuwei Wang[1], Gan Yang[1], Yiliang Liu[1], Yueyang Li[1], Yiqun Lu[1], Lihong Wang[1], Lin Wang[1,2,3,4,5]**

[1] Shanghai Key Laboratory of Atmospheric Particle Pollution and Prevention (LAP[3]), Department of Environmental Science & Engineering, Jiangwan Campus, Fudan University, Shanghai 200438, China

[2] Collaborative Innovation Center of Climate Change, Nanjing, 210023, China

[3] Shanghai Institute of Pollution Control and Ecological Security, Shanghai 200092, China

[4] IRDR International Center of Excellence on Risk Interconnectivity and Governance on Weather/Climate Extremes Impact
and Public Health, Fudan University, Shanghai 200438, China

[5] National Observations and Research Station for Wetland Ecosystems of the Yangtze Estuary, Shanghai, China

**Correspondence:** Lin Wang (lin_wang@fudan.edu.cn)

**ABSTRACT:** The volatility of organic aerosols plays a key role in determining their gas-particle partitioning, which subsequently alters the physicochemical properties and atmospheric fates of aerosol particles. Nevertheless, an accurate
estimation of the volatility of organic aerosols (OA) remains challenging, because most standards for particulate organic compounds are not available, and even for those with standards, their vapor pressures are too low to be measured by most traditional methods. Here, we deployed an iodide-adduct Long Time-of-Flight Chemical Ionization Mass Spectrometer (LToF-CIMS) coupled with a Filter Inlet for Gases and AEROsols (FIGAERO) to probe the relationship between the molecular formulae of atmospheric organic aerosols' components and their volatilities. $T_{max}$ (*i.e.*, the temperature corresponding to the
first signal peak of thermogram) for calibrants were abstracted and validated from the desorption thermograms of mixed organic and inorganic calibrants that were atomized and then collected onto a PTFE filter, leading to a linear correlation between $T_{max}$ and volatility. In addition, 30 ambient filter samples were collected in winter 2019 at Wangdu station in Beijing-Tianjin-Hebei region, and analyzed by FIGAERO-LToF-CIMS, leading to the identification of 1,448 compounds dominated by the CHO (containing carbon, hydrogen, and oxygen atoms) and CHON (containing carbon, hydrogen, oxygen, and nitrogen
atoms) species. Among them, 181 organic formulae including 91 CHO and 90 CHON compounds were then selected since their thermograms can be characterized with clear $T_{max}$ values in more than 20 out of 30 filter samples, and subsequently divided into two groups according to their O/C ratios and different thermal desorption behavior. The mean O/C of these two groups are $0.56 \pm 0.35$ (average ± one standard deviation) and $0.18 \pm 0.08$, respectively. Then the parameterizations between volatility and elemental composition for the two group compounds were obtained. Compared with previous volatility
parameterizations, our functions provide a better estimation for the volatility of low-volatility organic compounds (LVOCs) in ambient organic aerosols. Furthermore, our results suggest that volatility parameterizations should be specialized for organic compounds with different O/C ratios.

## 1 Introduction

Aerosol particles can significantly impact human health, visibility and climate (Stocker et al., 2013). Organic aerosol (OA)
comprises tens of thousands of organic substances and makes up a significant mass fraction of the total submicron particles in the troposphere (Jimenez et al., 2009). Whether an organic compound will exist in the gas phase or particles under a specific temperature is determined by its volatility, which depends on its molar mass and functional groups (Capouet and Müller, 2006; Pankow and Asher, 2008). The volatility of a compound is usually expressed as saturation mass concentration ($C_0$) or saturation vapor pressure ($Psat$). The effective saturation mass concentration ($C^*$) includes the effect of non-ideal thermodynamic mixing
with an activity coefficient ($\gamma$), thus $C^* = \gamma C_0$, and $C^*$ equals $C_0$ under the assumption of ideal thermodynamic mixing (Donahue et al., 2011). Saturation mass concentration is regarded as one of critical physicochemical parameters for organic

aerosols' components. The organic compounds with $C^* < 0.1$ $\mu g \cdot m^{-3}$ are mostly in the condensed phase, the organic compounds with $C^* > 1000$ $\mu g \cdot m^{-3}$ are almost entirely in the gas phase, and the organic compounds with $1 < C^* < 100$ $\mu g \cdot m^{-3}$ will be found in both phases under typical conditions (Donahue et al., 2009).


During the past years, two major methods relevant to the Chemical Ionization Mass Spectrometer (CIMS) have been developed to characterize the volatility of aerosols. The first one estimates the volatility of an organic species based on its molecular formula. The relationship between $C^*$ and molecular formulae of alkane, aldehyde, ketone, alcohol, acid, diol, and diacid was proposed by Donahue et al. (2011), which clarifies the relationship between $n_C$ (the numbers of carbon) and $n_O$ (the numbers of oxygen), and log$C_0$. The relationship was derived from a group contribution method SIMPOL that actually is a structure-based estimation method (Pankow and Asher, 2008). Li et al. (2016) updated this function by including 31066 compounds from the National Cancer Institute (NCI) open database, which applies to not only CHO compounds (containing carbon, hydrogen, and oxygen atoms) but also the nitrogen- and sulfur-containing compounds. However, Isaacman-Vanwertz and Aumont. (2021) showed that the volatility of CHON compounds estimated by the Li et al. (2016) parameterization is significantly biased by an increase in the number of nitrogen atoms, and thus they modified the nitrogen coefficient for CHON compounds from Li et al. (2016) study by using a fixed relationship between the nitrogen coefficient and the number of the oxygen atom (*i.e.*, $b_N=-2*b_O$). On the other hand, Donahue et al. ( 2011) took only -OH, =O and -C(O)OH functionalities into account when describing an average effect of an added oxygen, which could result in a large uncertainty when estimating the volatility of highly oxygenated organic molecules (HOMs) that contain hydroperoxide (-OOH) functionalities. Thus, Stolzenburg et al. (2018) and Mohr et al. (2019) updated the parameters for the volatility estimation of HOMs, based on 15 HOMs. The molecular structures of these 15 HOMs are unclear, but their saturation concentrations were estimated using the SIMPOL method on the basis of supposed molecular structures (Tröstl et al., 2016). As the covalently bonded dimers are abundant in HOMs from ozonolysis of $\alpha$-pinene, Stolzenburg et al. (2018) fitted parameters using monomer and dimer HOMs separately, allowing a parameter to represent the covalent binding. Since molecular formulae of organic aerosols can be obtained by state-of-the-art instruments such as high-resolution mass spectrometers, $C^*$ of organic compounds in the aerosol particles can then be calculated based on the above-mentioned empirical functions (Huang et al., 2019).

The second one estimates the volatility of an organic species on the basis of its desorption thermogram. When analyzing physicochemical properties of aerosol particles, one of the most popular techniques is to heat the particles and then detect the evaporated compounds utilizing mass spectrometry techniques, such as Thermodenuder-particle Beam Mass Spectrometer (Faulhaber et al., 2009), Thermal-Desorption Chemical Ionization Mass Spectrometer (TD-CIMS) (Smith et al., 2004), Micro-Orifice Volatilization Impactor Coupled to a Chemical Ionization Mass Spectrometer (MOVI-CIMS) (Yatavelli and Thornton, 2010), Chemical Analysis of Aerosols Online -Proton Transfer Reaction Mass Spectrometer (CHARON-PTR-MS) (Eichler et al., 2015), and the Filter Inlet for Gases and AEROsols (FIGAERO) coupled with Time-of-Flight Chemical Ionization Mass Spectrometer (ToF-CIMS) (Lopez-Hilfiker et al., 2014). Basically, the desorption temperature is ramped up linearly, the particulate organic compounds with different vapor pressures are thermo-desorbed and then characterized with distinct thermograms (*i.e.,* desorption signal versus temperature), and the temperature corresponding to the first peak signal ($T_{max}$) correlates with the vaporization enthalpy of a compound (Lopez-Hilfiker et al., 2014). It is thus applicable to estimate $C^*$, *i.e.,* the volatility of the chemical constituents in the particles, from the measured $T_{max}$, after calibration with a set of standards with known vapor pressures (Bannan et al., 2019), which has been widely applied in many previous studies (Nah et al., 2019; Stark et al., 2017; Wang et al., 2020a; Ye et al., 2019; Ylisirniö et al., 2019). Compared with the parameterization method from organic aerosols' molecular formulae, the thermogram method is able to give a volatility distribution that is likely closer to the real one. The molecular formula method likely treats the thermal decomposition products after heating as evaporated organic molecules, and thus overestimates the overall volatility of a group of organics (Stark et al., 2017).

85

The FIGAERO-ToF-CIMS has been widely used in the field and laboratory studies in recent years. For example, Wang et al. (2020a) explored the volatility of aromatic hydrocarbon photo-oxidation products, Ylisirniö et al. (2020) compared the volatility of SOA (Secondary Organic Aerosol) components formed from oxidation of real tree emissions with that formed

from oxidation of single VOC-systems, and Ye et al. (2019) studied the volatility of nucleated particles from α-Pinene oxidation between -50 ˚C and +25 ˚C using FIGAERO-ToF-CIMS. Accurately measuring the desorption thermograms of the standards is one of the essential factors for the success of this method. For example, previous studies typically used the syringe deposition method to prepare the mimic filter, which leads to wide variations in results. A new method for volatility calibration, the atomization method, accurately captures the evaporation of chemical constituents from ambient aerosol particles (Ylisirniö et al., 2021). In addition, the influences of mixing of organic compounds and inorganic salts that are a major component in ambient aerosol particles on the thermograms of organics, were not considered in these previous studies.

On the other hand, with rapid economic growth and urbanization in the North China Plain (NCP), air pollution and extreme haze events frequently occurred in this region, the formation of which is closely related to the volatility of aerosol components (Li et al., 2017; Shiraiwa and Seinfeld, 2012). Therefore, it is crucial to understand the volatility of aerosol components in the NCP, and to a larger extent, in the ambient atmosphere.

In this study, we compared the effects of the methods of syringe deposition and atomization on $T_{max}$ with a series of authentic organic standards using FIGAERO-LToF-CIMS and investigated the influences of inorganic salts and the mixing of organic compounds on the $T_{max}$ of organics. In addition, we developed empirical volatility-molecular formula functions based on selected CHO and CHON compounds with varying O/C ratios from ambient particles collected at Wangdu station in the North China Plain, China, from January 15 to 22, 2019. The $C*$ of these selected compounds were estimated by obtained $T_{max}$ from thermograms. Lastly, our empirical functions were compared with previous ones.

## 2 Experimental methods

### 2.1 FIGAERO-LToF-CIMS

The chemical composition and thermograms of particulate compounds collected on filters were measured via a FIGAERO-LToF-CIMS with a mass resolving power of 7700-8500, and the volatility of compounds was acquired from thermograms (Bertram et al., 2011; Lee et al., 2014; Lopez-Hilfiker et al., 2014). The design and operation of the FIGAERO have been introduced in previous studies (Bannan et al., 2019; Lopez-Hilfiker et al., 2014; Thornton et al., 2020; Ye et al., 2020). In this study, particles collected in the lab calibration experiments or from the field campaign were thermally desorbed utilizing an ultrahigh purity (UHP) nitrogen flow at 2.3 Liter per minute$^{-1}$ (lpm), among which 1.0 lpm UHP $N_2$ passed the filter and entered the ion-molecule reaction (IMR) chamber. In IMR, organic molecules were charged by iodide ions generated by exposure of a 1.0 lpm mixture of $CH_3I$ and UHP $N_2$ to a 0.1 mCi radioactive Am-241 source.

The desorption procedure for the calibration experiments and the field measurements were the same, as shown in Figure S1. During the thermo-desorption process, the heating temperature ramp linearly started from room temperature (~25 ˚C) to 134 ˚C and then the filter was held at 134 °C for 40 min to ensure that most of the organic compounds were desorbed from the filter (Lopez-Hilfiker et al., 2016). The measured ramping rate for heating was 2.27 ˚C /min in this study. A slower ramping rate allows more time to stay at any momentary desorption temperature so that a larger fraction of molecules would evaporate (Ylisirniö et al., 2021). Also, a slower ramping rate can separate compounds with similar volatilities better (Lopez-Hilfiker et al., 2014). Note that the decomposition degree of parental compounds under a slower ramping rate is higher than that under a faster ramping rate, but a slower heating rate leads to a smaller number of thermal decomposition products (Yang et al., 2021). Most of the ambient organic compounds can be desorbed from the filter at less than 134 °C (Huang et al., 2019). Furthermore, high molecular weight organic compounds (e.g. $C_{27}H_{52}O_4$ ) can be evaporated from the filter below 120 °C (Wang et al., 2016; Zhang et al., 2020). Therefore, the highest temperature of 134 ˚C is feasible in our study.

During ambient filter measurements, background measurements using a blank filter were also conducted. The blank filter was analyzed by the same thermal desorption procedure as that of the field samples. The obtained signals are treated as the background signals. An example of the background signal of an identified compound was shown in Figure S2.

A Tofware software (version 3.1.2, Tofwerk AG, Switzerland) was used to analyze the data of the mass spectrometer. To plot thermograms, signals of evaporated compounds were normalized by the abundance of reagent ions and then subtracted with the background signals, which were normalized similarly. The raw data was acquired at a frequency of 1 Hz and then averaged to a 20 s time interval during the data analysis. As the desorption features of ambient aerosol particles were quite complex, we applied the Levenberg–Marquardt algorithm to fit the thermograms and conducted peak deconvolution for the multimodal

thermograms (Goodman and Brenna, 1994; Lopez-Hilfiker et al., 2015; Stolzenburg et al., 2018). In the case of a multimodal thermogram, the higher-temperature peak(s) (*i.e.,* the warmer peak ) was assumed to come from the thermal decomposition of larger molecules or isomers with different vapor pressures (Huang et al., 2018; Wang et al., 2016; Zhang et al., 2020). Hence, in this study, $T_{max}$ of the cooler peak (*i.e.,* the first peak) was used to estimate the volatility of an organic compound.

**2.2 Calibration experiments**

The $C^*$ versus $T_{max}$ calibration curve was obtained by species with known vapor pressures. Two methods referred as the syringe deposition method and the atomization method, were used to prepare filter samples of authentic compounds. For the syringe deposition method, certain amounts of authentic species dissolved in the acetonitrile solvent were injected onto a PTFE filter by a syringe. While the acetonitrile solvent was supposed to quickly evaporate from the filter and have a minor effect on authentic species, only the authentic species were thermally desorbed during the subsequent FIGAERO-LToF-CIMS analysis.

For the atomization method, the authentic species dissolved in deionized water were atomized by a commercial atomizer (TSI® 3076). Atomized particles were diluted and dried by a zero-gas flow and silica gel, respectively, after which the relative humidity (RH) of the flow was regulated to around ~2% (Figure S3). Then, particles were collected onto a PTFE filter and subsequently analyzed by the FIGAERO-LToF-CIMS. The mass of collected particles can be calculated based on the number size distribution of particles measured by a Scanning Mobility Particle Sizer (SMPS, TSI® 3776), the particle density, the

collection time, and the flow rate through the filter (Ylisirniö et al., 2021). Among them, the particle density can be estimated according to the density of corresponding authentic standards and their mixing ratios in the solution.

During our laboratory tests, eleven sets of calibration experiments were conducted. These experiment conditions are summarized in Table 1. One set of (No.1) experiments using the syringe deposition method were performed, where

polyethylene glycols (PEGs) were used as authentic organic standards (Bannan et al., 2019). Besides, ten sets of atomization experiments were also conducted. The No.2 set works as an intercomparison with previous syringe deposition (No.1) experiments. The No.3-7 sets of experiments were conducted to explore the effects of ammonium sulfate and mixed organic compounds on the $T_{max}$ of organics. Erythritol, PEG-6, PEG-7, PEG-8, and citric acid were used as authentic organic standards because they can co-dissolve with ammonium sulfate in deionized water. In No.3, ammonium sulfate was not added to the

standard solution, and there were only organic standards. In No.4 and No.5, ammonium sulfate was mixed with erythritol, PEG-6, PEG-7, PEG-8, and citric acid, respectively. 200 ng and 1000 ng atomized particles were collected in No.4 and No.5, respectively. It is assumed that the atomized particles were internally mixed with the same mass ratio as that in the solution (Drisdell et al., 2009), consisting of 100 ng/500 ng of ammonium sulfate and 100 ng/500 ng of the mixed organic standard. Although ammonium sulfate is much less volatile than mixed organics and the mixing ratio of ammonium sulfate to organic

compounds in atomized particles might be different with that in solution, this assumption likely leads to a minor effect on the matrix effect of ammonium sulfate. In the sixth (No.6) set, erythritol, PEG-6, PEG-7, PEG-8, and citric acid were mixed together and 1000 ng atomized particles were collected. In No.7, ammonium sulfate was mixed with erythritol, PEG-6, PEG-7, PEG-8, and citric acid. The No.7-11 sets of experiments were conducted to explore the effect of mass loading of filters. Four replicates were performed for each set of experiments.


**Table 1.** Conditions of eleven sets of calibration experiments.

| No. | Method | Authentic standards | Concentration | Solvent | Deposited volume | Mass loading |
|-----|--------|---------------------|---------------|---------|------------------|--------------|
| 1 | Syringe deposition | PEG-4 ($C_8H_{18}O_5$)<br>PEG-5 ($C_{10}H_{22}O_6$)<br>PEG-6 ($C_{12}H_{26}O_7$)<br>PEG-7 ($C_{14}H_{30}O_8$)<br>PEG-8 ($C_{16}H_{34}O_9$) | $0.05$ g $L^{-1}$<br>$0.05$ g $L^{-1}$<br>$0.05$ g $L^{-1}$<br>$0.05$ g $L^{-1}$<br>$0.05$ g $L^{-1}$ | Acetonitrile | 2 µl | 100 ng<br>100 ng<br>100 ng<br>100 ng<br>100 ng |
| 2 | Atomization | PEG-4 ($C_8H_{18}O_5$)<br>PEG-5 ($C_{10}H_{22}O_6$)<br>PEG-6 ($C_{12}H_{26}O_7$)<br>PEG-7 ($C_{14}H_{30}O_8$)<br>PEG-8 ($C_{16}H_{34}O_9$) | $1.0$ g $L^{-1}$<br>$1.0$ g $L^{-1}$<br>$1.0$ g $L^{-1}$<br>$1.0$ g $L^{-1}$<br>$1.0$ g $L^{-1}$ | Deionized water | / | 500 ng<br>500 ng<br>500 ng<br>500 ng<br>500 ng |
| 3 | Atomization | Erythritol ($C_4H_{10}O_4$)<br>PEG-6 ($C_{12}H_{26}O_7$)<br>PEG-7 ($C_{14}H_{30}O_8$)<br>PEG-8 ($C_{16}H_{34}O_9$)<br>Citric acid ($C_6H_8O_7$) | $0.5$ g $L^{-1}$<br>$0.5$ g $L^{-1}$<br>$0.5$ g $L^{-1}$<br>$0.5$ g $L^{-1}$<br>$0.5$ g $L^{-1}$ | Deionized water | / | 100 ng<br>100 ng<br>100 ng<br>100 ng<br>100 ng |
| 4 | Atomization | Erythritol ($C_4H_{10}O_4$) +ammonium sulfate<br>PEG-6 ($C_{12}H_{26}O_7$) + ammonium sulfate<br>PEG-7 ($C_{14}H_{30}O_8$) + ammonium sulfate<br>PEG-8 ($C_{16}H_{34}O_9$) + ammonium sulfate<br>Citric acid ($C_6H_8O_7$) + ammonium sulfate | $0.5$ g $L^{-1}$ + $0.5$ g $L^{-1}$<br>$0.5$ g $L^{-1}$ + $0.5$ g $L^{-1}$<br>$0.5$ g $L^{-1}$ + $0.5$ g $L^{-1}$<br>$0.5$ g $L^{-1}$ + $0.5$ g $L^{-1}$<br>$0.5$ g $L^{-1}$ + $0.5$ g $L^{-1}$ | Deionized water | / | 200 ng (100 ng + 100 ng)<br>200 ng (100 ng + 100 ng)<br>200 ng (100 ng + 100 ng)<br>200 ng (100 ng + 100 ng)<br>200 ng (100 ng + 100 ng) |
| 5 | Atomization | Erythritol ($C_4H_{10}O_4$) +ammonium sulfate<br>PEG-6 ($C_{12}H_{26}O_7$) + ammonium sulfate<br>PEG-7 ($C_{14}H_{30}O_8$) + ammonium sulfate<br>PEG-8 ($C_{16}H_{34}O_9$) + ammonium sulfate<br>Citric acid ($C_6H_8O_7$) + ammonium sulfate | $0.5$ g $L^{-1}$ + $0.5$ g $L^{-1}$<br>$0.5$ g $L^{-1}$ + $0.5$ g $L^{-1}$<br>$0.5$ g $L^{-1}$ + $0.5$ g $L^{-1}$<br>$0.5$ g $L^{-1}$ + $0.5$ g $L^{-1}$<br>$0.5$ g $L^{-1}$ + $0.5$ g $L^{-1}$ | Deionized water | / | 1000 ng (500 ng + 500 ng)<br>1000 ng (500 ng + 500 ng)<br>1000 ng (500 ng + 500 ng)<br>1000 ng (500 ng + 500 ng)<br>1000 ng (500 ng + 500 ng) |
| 6 | Atomization | Erythritol ($C_4H_{10}O_4$) + PEG-6 ($C_{12}H_{26}O_7$) +PEG-7 ($C_{14}H_{30}O_8$) + PEG-8 ($C_{16}H_{34}O_9$) + citric acid ($C_6H_8O_7$) | $0.5$ g $L^{-1}$ + $0.5$ g $L^{-1}$ +$0.5$ g $L^{-1}$ + $0.5$ g $L^{-1}$ +$0.5$ g $L^{-1}$ | Deionized water | / | 1000 ng (200 ng + 200 ng +200 ng +200 ng + 200 ng) |
| 7 | Atomization | Erythritol ($C_4H_{10}O_4$) + PEG-6 ($C_{12}H_{26}O_7$) + PEG-7 ($C_{14}H_{30}O_8$) + PEG-8 ($C_{16}H_{34}O_9$) + citric acid ($C_6H_8O_7$) + ammonium sulfate | $0.5$ g $L^{-1}$ + $0.5$ g $L^{-1}$ +$0.5$ g $L^{-1}$ + $0.5$ g $L^{-1}$ +$0.5$ g $L^{-1}$ + $2.5$ g $L^{-1}$ | Deionized water | / | 1000 ng (100 ng + 100 ng +100 ng +100 ng + 100 ng + 500 ng) |
| 8 | Atomization | Erythritol ($C_4H_{10}O_4$) + PEG-6 ($C_{12}H_{26}O_7$) + PEG-7 ($C_{14}H_{30}O_8$) + PEG-8 ($C_{16}H_{34}O_9$) + citric acid ($C_6H_8O_7$) + ammonium sulfate | $0.5$ g $L^{-1}$ + $0.5$ g $L^{-1}$ +$0.5$ g $L^{-1}$ + $0.5$ g $L^{-1}$ +$0.5$ g $L^{-1}$ + $2.5$ g $L^{-1}$ | Deionized water | / | 200 ng (20 ng + 20 ng +20 ng +20 ng + 20 ng + 100 ng) |
| 9 | Atomization | Erythritol ($C_4H_{10}O_4$) + PEG-6 ($C_{12}H_{26}O_7$) + PEG-7 ($C_{14}H_{30}O_8$) + PEG-8 ($C_{16}H_{34}O_9$) + citric acid ($C_6H_8O_7$) + ammonium sulfate | $0.5$ g $L^{-1}$ + $0.5$ g $L^{-1}$ +$0.5$ g $L^{-1}$ + $0.5$ g $L^{-1}$ +$0.5$ g $L^{-1}$ + $2.5$ g $L^{-1}$ | Deionized water | / | 500 ng (50 ng + 50 ng +50 ng +50 ng + 50 ng + 250 ng) |
| 10 | Atomization | Erythritol ($C_4H_{10}O_4$) + PEG-6 ($C_{12}H_{26}O_7$) + PEG-7 ($C_{14}H_{30}O_8$) + PEG-8 ($C_{16}H_{34}O_9$) + citric acid ($C_6H_8O_7$) + ammonium sulfate | $0.5$ g $L^{-1}$ + $0.5$ g $L^{-1}$ +$0.5$ g $L^{-1}$ + $0.5$ g $L^{-1}$ +$0.5$ g $L^{-1}$ + $2.5$ g $L^{-1}$ | Deionized water | / | 1500 ng (150 ng + 150 ng +150 ng +150 ng + 150 ng + 750 ng) |
| 11 | Atomization | Erythritol ($C_4H_{10}O_4$) + PEG-6 ($C_{12}H_{26}O_7$) + PEG-7 ($C_{14}H_{30}O_8$) + PEG-8 ($C_{16}H_{34}O_9$) + citric acid ($C_6H_8O_7$) + ammonium sulfate | $0.5$ g $L^{-1}$ + $0.5$ g $L^{-1}$ +$0.5$ g $L^{-1}$ + $0.5$ g $L^{-1}$ +$0.5$ g $L^{-1}$ + $2.5$ g $L^{-1}$ | Deionized water | / | 2000 ng (200 ng + 200 ng +200 ng +200 ng + 200 ng + 1000 ng) |

## 2.3 Field Campaign

An ambient campaign was conducted from December 16, 2018 to January 22, 2019 at Wangdu station, Hebei Province, China (Hu et al., 2022; Wang et al., 2020b). The campaign site (38.66 °N, 115.19 °E) was mainly influenced by the surrounding 185 transportation, industrial and residential sources, and farmlands and forests and can be treated as a typical suburban station.

Aerosol particles ($PM_{2.5}$) were collected four times every day, and each collection lasted for 15 minutes (*i.e.,* 7:00-7:15, 12:00-12:15, 17:30-17:45, and 21:00-21:15 local time, respectively). Ambient $PM_{2.5}$ was sampled onto PTFE filters (5 μm pore size, 25 mm diameter, Millipore), and the flow rate was regulated at 1.42 L min$^{-1}$. After the collection, filter samples were preserved at -20 ˚C in a freezer until further analysis. In this study, 30 filter samples between January 15, 2019 and January 22, 2019 were analyzed with FIGAERO offline, because mass loadings of these 30 filter samples varied from 200 ng to 3500 ng with a median of 1100 ng, which is similar to those in the calibration experiments. The mass concentration of $PM_{2.5}$ was measured by a commercial synchronized hybrid real-time particulate monitor (TEI, Model 5030i).

## 2.4 Saturation mass concentration ($C$*)

### 2.4.1 Calculation of $C$*

By correlating the logarithm of $P_{sat}$ at 298K of these authentic standards in the literature to their $T_{max}$ values obtained from the desorption thermograms, a linear relationship can be obtained (Bannan et al., 2019):

$$log_{10}(P_{sat}) = aT_{max} + b, \qquad (1)$$

where $a$ and $b$ are fitted parameters, and this expression can also be expressed as:

$$P_{Sat}(pa) = 10^{aT_{max}+b}, \qquad (2)$$

On the other hand, $P_{sat}$ can be converted to $C$* with the assumption of the ideal gas law (Ylisirniö et al., 2020, 2021). In this way, the relationship between $C$* and $T_{max}$ is deduced as:

$$C^* (\mu g \cdot m^{-3}) = \frac{(10^{aT_{max}+b})M_w}{RT} 10^6, \qquad (3)$$

where $M_w$ is the molecular weight of an authentic compound (g mol$^{-1}$), $R$ is the gas constant (8.314 J mol$^{-1}$ K$^{-1}$), and $T$ is the temperature when the $P_{sat}$ is determined (K; in our study, $T$ is 298 K).

### 2.4.2 Correlation between $C$* and molecular formulae

We substituted the measured $T_{max}$ of selected organic compounds in the ambient aerosol particles into Eq. (3) with fitted $a$ and $b$ values from experiments with authentic standards, and obtained their $C$*. Then we correlated $C$* to molecular formulae of these selected organic compounds in a function similar to what has been developed in a previous study (Donahue et al., 2011; Mohr et al., 2019):

$$log_{10} C_0 = (n_c^0 - n_c)b_c - (n_o - 3n_N)b_o - 2 \cdot \frac{(n_o - 3n_N)n_c}{(n_c + n_o - 3n_N)} b_{co} - n_N b_N \qquad (4)$$

where $n_c^0$ is the reference carbon number and set to be 25 (Donahue et al., 2011); $n_c$, $n_o$ and $n_N$ is numbers of carbon, oxygen and nitrogen atoms in an organic species, respectively; $b_c$, $b_o$ and $b_N$ denote the contribution of each kind of atoms to $log_{10} C_0$, respectively, and $b_{co}$ is the carbon-oxygen nonideality (Donahue et al., 2011; Li et al., 2016). Values of $b_c$, $b_o$, $b_N$, and $b_{co}$ are then fitted with multi-linear least-squares analysis.

## 3 Results and discussion

### 3.1 Laboratory Calibration

Figure S4 compares the $T_{max}$ values for the same authentic organic standards when using different calibration methods. PEG-4 was not detected by CIMS with the second (No.2) set of calibration experiments, which is consistent with the result of a previous study (Ylisirniö et al., 2021). This observation is most likely due to the high volatility of PEG-4 that leads to its evaporation even before the CIMS measurement (Ylisirniö et al., 2021). The $T_{max}$ values measured with the syringe deposition experiments have larger error bars (Figure S4). The $T_{max}$ may increase with increased filter loadings (Wang and Hildebrandt Ruiz, 2018), if calibrated with the same method. However, despite a larger mass loading (500 ng) in the atomization (No.2) experiments than that (100 ng) in the syringe deposition (No.1) experiments, the $T_{max}$ values measured with the atomization

method is about 20 ˚C lower than those with the syringe deposition method for the same compound. This observation can be explained by the fact that the surface area of the material deposited by the syringe is smaller than that of deposited aerosol particles, which requires more time to evaporate and corresponds to higher $T_{max}$ values (Ylisirniö et al., 2021).

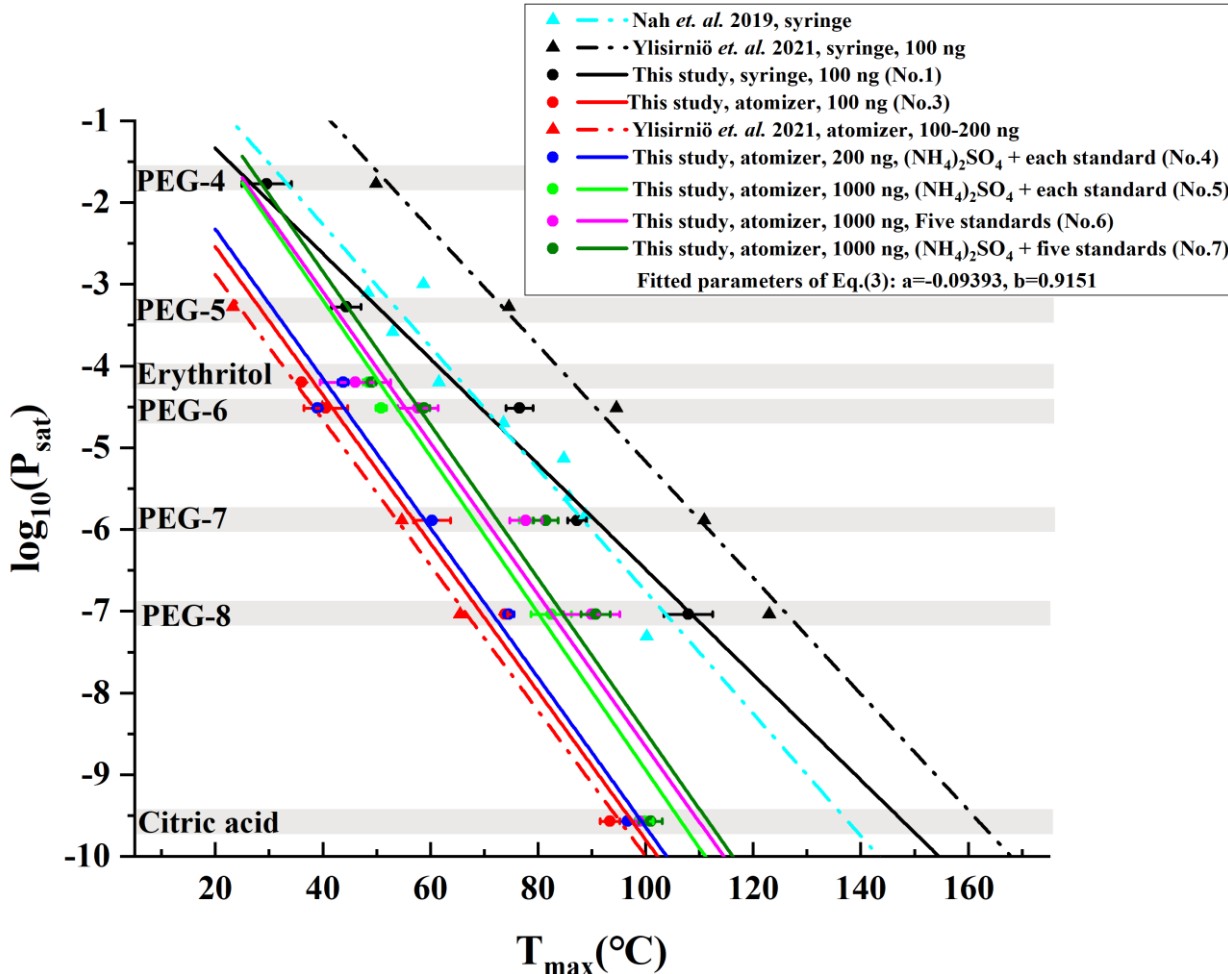


**Figure 1.** Comparison of calibration results obtained in this study with those reported previously. These solid lines denote the calibration results obtained in this study. Error bars represent ± one standard deviation of $T_{max}$ from four replicate experiments. The fitted parameters correspond to the dark green line. The light blue dash-dot line denotes calibration results obtained with acids and erythritol by Nah et al. (2019) using the syringe method. The black dash-dot line represents calibration results

obtained with 100 ng deposited PEGs (including PEG-4, PEG-5, PEG-6, PEG-7, and PEG-8) by Ylisirniö et al. (2021) using the syringe method. The red dash-dot line denotes calibration results obtained with 100-200 ng PEGs (including PEG-5, PEG-6, PEG-7, and PEG-8) by Ylisirniö et al. (2021) using the atomization method.

In Figure 1, we further compared our calibration results with previously reported ones. Six solid calibration lines acquired in

this study are located between the dash-dot calibration lines by Ylisirniö et al. (2021) using the atomization method and the syringe deposition method. The solid calibration line obtained with PEGs (O/C > 0.25 ) by the syringe deposition method in this study is quite close to the dash-dot calibration line that was also obtained by the syringe deposition method with acids (O/C: 0.09-0.80 ) and erythritol (O/C > 0.25 ) (Nah et al., 2019). Yet the slopes of two lines are a bit different, which may be attributed to different O/C ratio of organic standards. In addition, the calibration line obtained with 100 ng deposited standards

in this study by the atomization method almost overlaps that obtained with the same method, standard, and mass loading (Ylisirniö et al., 2021). However, the calibration line obtained with 100 ng deposited standards in this study by the syringe deposition method is far away from the dash-dot calibration line obtained with the same method, standard, and mass loading

(Ylisirniö et al., 2021). Clearly, compared with the syringe deposition method, the atomization method shows much better repeatability, even between different studies.


The effects of ammonium sulfate and mixed organic compounds on $T_{max}$ of organics are also investigated, as shown in Figure 1 and Figure S5, since the majority of current atmospheric aerosol particles consists of ~50% ammonium sulfate and 50% carbonaceous components (Drisdell et al., 2009). Clearly, mixing the same amount of ammonium sulfate with each of the five organic standards, as done in No.4, increased the $T_{max}$ values of erythritol and citric acid but did not alter the $T_{max}$ values of

PEGs 6-8 significantly. Comparison of the $T_{max}$ values of No.6 and No.7 shows that the $T_{max}$ of five organics exhibited a slight increase (1 to 4 ℃), which is likely due to the addition of ammonium sulfate. Furthermore, elevated $T_{max}$ (1 to 8 ℃) between No.5 and No.7 was observed, which means the matrix effects within mixed organic compounds can also enhance the $T_{max}$ of organics. The $T_{max}$ of five organic compounds increased by 3 to 17 ℃ with the increase of mass loadings according to No.4 and No.5 (Figure S5). Furthermore, as shown in Figure S6, the $T_{max}$ of five organic compounds generally increases with

increased mass loadings, and $T_{max}$ has increased approximately 8 ℃ as the mass loading increases from 200 ng to 1500 ng. The 95% credible intervals of No.5, No.6 and No.7 experiments are significantly larger than the others, which may be attributed to their higher mass loading (1000 ng) than those in other experiments (100 ng, 200 ng and 500 ng) (Figure S7). Therefore, the $T_{max}$ values of organic compounds are affected together by the addition of ammonium sulfate, the matrix effects within organic compounds, and mass loadings. However, these effects cannot be quantified separately in our study.


Moreover, the fraction of organic species and inorganic salts of non-refractory submicron aerosol species (NR-PM1) in Beijing in winter 2018 were about 48% and 52%, respectively (Zhou et al., 2020). The fraction of organic species and inorganic salts in total particulate matters (PM) in a rural site (Gucheng in Hebei province) in winter 2018 were about 40% and 60%, respectively (Xu et al., 2021). The mass ratios of the inorganic salt to organic species were close to 1:1 which was similar to

that of our laboratory tests (*i.e.,* No.7 set of calibration experiments). In addition, Ylisirniö et al. (2021) shows that particle size has a moderate impact on the measured $T_{max}$ of organic compounds. The particle size distributions and peak diameters of polydisperse particles in our laboratory experiments (No.4, No.6 and No.7) are similar to those of the ambient samples (Figure S8). Therefore, in our study, particle size distributions have a minor effect on measured $T_{max}$.

To minimize the uncertainties from multiple factors (e.g., the presence of ammonium sulfate, multiple organic compounds, particle size distributions, and mass loading) on $T_{max}$, the calibration line obtained from No.7 was utilized to estimate $T_{max}$ values of organic compounds in ambient particles and to derive our parameterizations, because the experimental conditions of mimic particle samples in No.7 are the closest to those of the ambient samples and can represent ambient organic aerosol particles.

**3.2 Volatility of OA components**

We identified 1,448 compounds from the filter collected on 7:00-7:15, January 15, 2019, in Wangdu, whose mass defect plot is shown in Figure 2. Among them, 340 CHO and 663 CHON species account for 43.5% and 20.8% of the total signals, respectively, because the iodide-adduct chemical ionization is sensitive toward multifunctional oxygenated organic compounds with minimal fragmentation (Bertram et al., 2011; Lopez-Hilfiker et al., 2016). In addition to 326 other species (30.8% of the

total signals) that have been assigned with molecular formulae but cannot be divided into either the CHO or CHON groups, there are 119 species (4.9% of the total signals) without attributed molecular formulae.

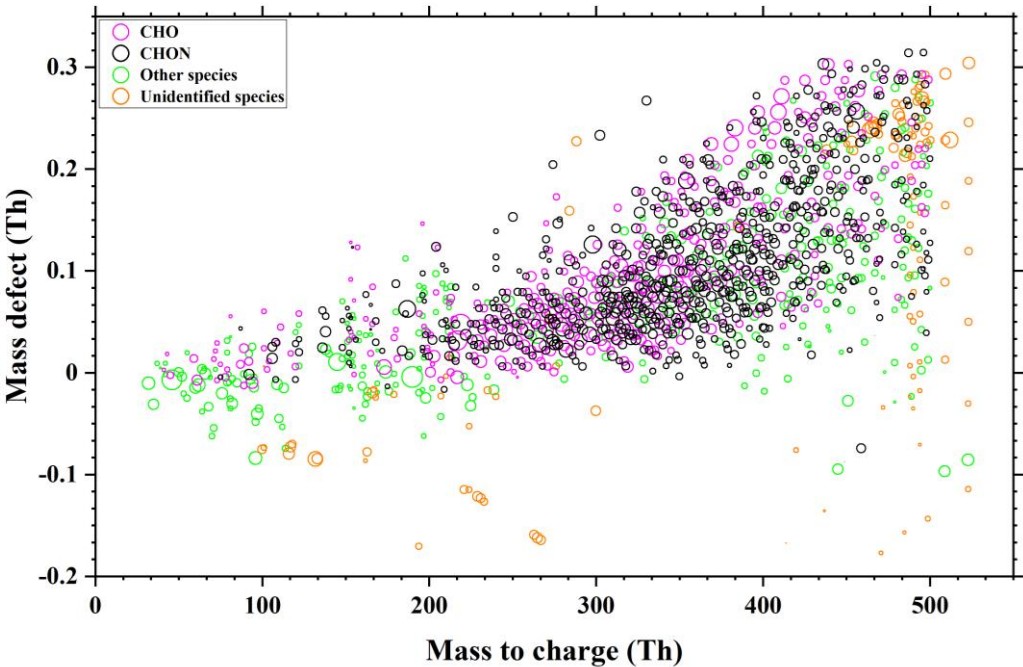

**Figure 2.** A typical mass defect plot for compounds desorbed from a filter collected on 7:00-7:15, January 15, 2019, in Wangdu. The symbol size is proportional to the logarithm of signal intensity. The reagent ion (I⁻) was not removed from their formulae.

Among the 1,448 species, we can attribute a $T_{max}$ to 765 species whose thermograms are characterized with a distinguishable $T_{max}$ with one or two desorption peaks, as shown in Figure S9a and c and Figure S9b and d, respectively. For the rest, their thermograms did not show a peak and thus the position of $T_{max}$ cannot be judged, as shown in Figure S9e and f. In Figure S10, we show the thermal desorption temperature of these 765 particulate compounds during the FIGAERO-LToF-CIMS analysis. The desorption temperatures of these organic compounds concentrated in the 80~100 ˚C range. The thermograms of most organic compounds show a single peak, and the mass-to-charge ratio (m/z) of these compounds are concentrated in the range of 250~450 Th, and the dominated compounds are $C_{13}H_{25}NO_2$, $C_{16}H_{32}O_2$, $C_{18}H_{35}NO_4$, $C_6H_{10}O_5$, $C_9H_{17}NO_2$ and $C_{18}H_{34}O_2$.

We analyzed 30 filter samples in total. For each filter, we selected species that can be assigned with a reliable molecular formula in the format of either CHO or CHNO, and species that can be designated with a $T_{max}$, the intersection of which correspond to species with both reliable CHO/CHNO type molecular formulae and $T_{max}$ values. There are 181 such organic compounds including 91 CHO and 90 CHON species that were present in more than 20 out of 30 filter samples. The 181 species are dominant compounds accounting for 34.1% of the total signal of 1448 compounds. It should be noted that several compounds (e.g., $C_3H_8O_3$, $CH_2O_2$) with high signals in these 1448 compounds were not further analyzed, because they are very volatile at the room temperature (25 ˚C) and their thermograms cannot be characterized with clear $T_{max}$ values from most filter samples. The molecular formula, molecular weight, $T_{max}$, and $C^*$ calculated according to our calibration in Figure 1 for the 91 CHO and 90 CHON species are summarized in Table S1 and Table S2, respectively.

The thermal behaviors of these 181 organic compounds during the FIGAERO analysis are shown in Figure 3. The cooler peak temperatures in double-peak thermograms mostly appeared in the green rectangular band of 45~80 ˚C, whereas the higher peak temperatures in double-peaks those are mainly the result of thermal decomposition of higher molecular weight organic compounds (Huang et al., 2018), and concentrated in the purple rectangular band of 100~125 ˚C, which is consistent with the result of Wang et al. (2016). On the other hand, the corresponding evaporation temperature for compounds with single-peak thermograms concentrated in the red rectangular band of 80~100 ˚C. Clearly, the compounds in Figure 3 can be divided into two groups, as illustrated with the two dashed ellipses. For each group, the $T_{max}$ values of the single peaks and the cooler ones

of double-peaks increase with their corresponding molecular weight, which is consistent with the fact that similar compounds with larger molecular weight tend to possess lower volatility.

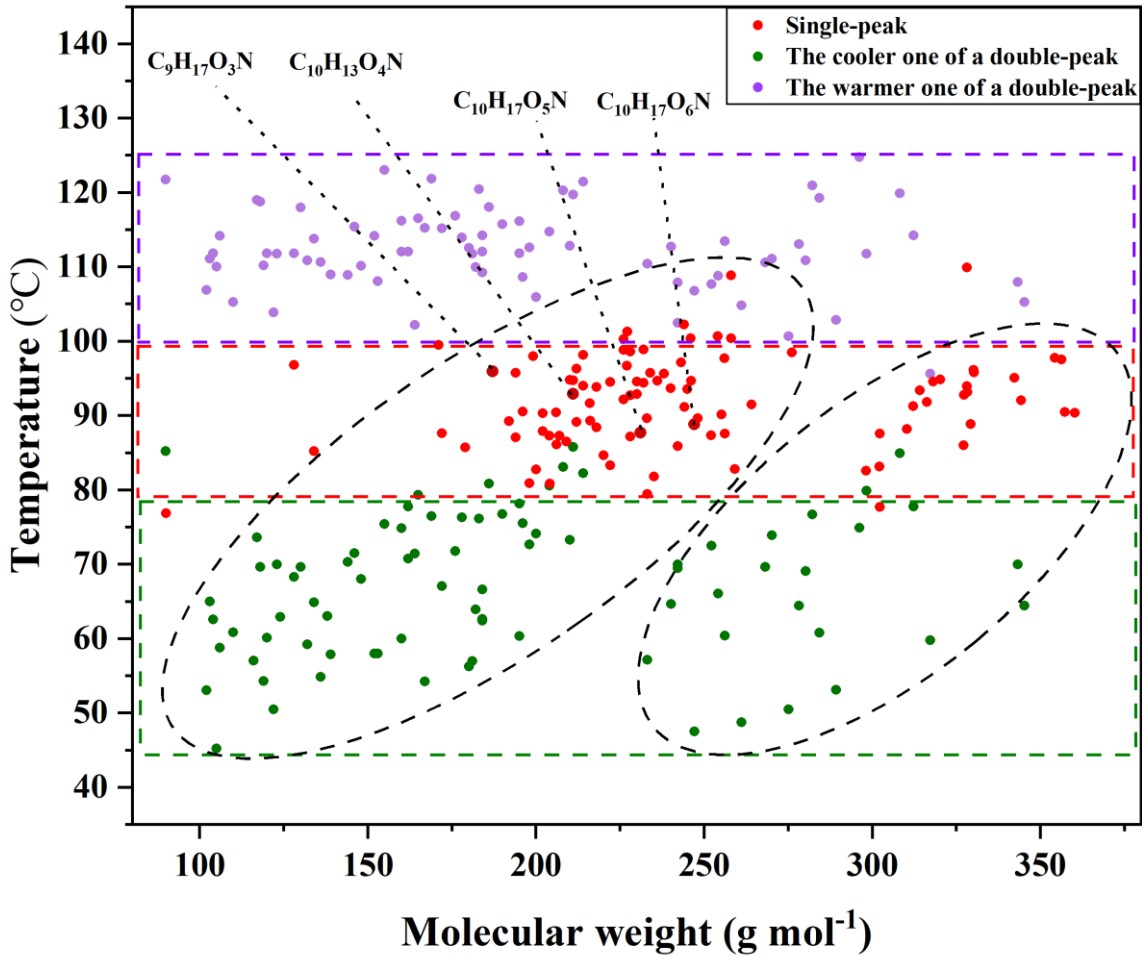


**Figure 3.** Evaporation and decomposition of 91 CHO and 90 CHON compounds, of which the reagent ion (I⁻) is excluded from their formulae. The signal peaks, the cooler and warmer peaks of double-peaks are denoted by red, green and purple circles, respectively. Rectangular bands depict the temperature zones in which peaks appear.

In Figure 4, the $T_{max}$ of these 181 compounds (91 CHO and 90 CHON) are translated into $C^*$ values according to Eq. (3). Since most of the warmer peaks of double peaks could be from thermal decomposition, the $T_{max}$ for the higher-temperature ones in double-peak thermograms of 78 compounds are not taken into account. Furthermore, thermal decomposition of the oligomers in organic aerosols can lead to a misinterpretation of the SOA volatility, and double bond equivalent (DBE) has been used to determine the thermal decomposition degree of an individual compound and SOAs formed from the ozonolysis of α-pinene

(Riva et al., 2019; Yang et al., 2021). The contents in Figure 4 are further colour-coded by DBE, instead of O:C, as shown in Figure S11. In Figure S11, the DBE distribution of these 181 compounds is random, thus the thermal decomposition could have a minor effect on $T_{max}$. On the other hand, 33 out of the 103 unimodal compounds meet the screening criteria of Yang et al. (2021) for considerable thermal decomposition (DBE ≥2, $T_{max}$ ≥72 °C and nO >4), and the unimodal thermograms of these 33 compounds generally do not present broad tailing and fronting, as shown in Figure S9a and c. The possible thermal

decomposition products of these 33 compounds are then investigated. Although thermal decomposition could be very complex, here we only considered dehydration products and decarboxylation products. $C_9H_{17}O_3N$ and $C_{10}H_{13}O_4N$ could be a decarboxylation product of $C_{10}H_{17}O_5N$ and a secondary dehydration product of $C_{10}H_{17}O_6N$, respectively. $T_{max}$ of $C_9H_{17}O_3N$ is higher than that of $C_{10}H_{17}O_5N$ by about 8 °C and $T_{max}$ of $C_{10}H_{13}O_4N$ is higher than that of $C_{10}H_{17}O_6N$ by about 4 °C, which is close to the results of Yang et al. (2021). However, this observation can also be explained by isomers with vastly different

vapor pressures (Huang et al., 2018; Lopez-Hilfiker et al., 2015). For a quick identification of these compounds, $C_9H_{17}O_3N$, $C_{10}H_{17}O_5N$, $C_{10}H_{13}O_4N$ and $C_{10}H_{17}O_6N$ have been marked in Figure 3. Since only two compounds may be thermal decomposition fragments, thermal decomposition likely has little effects on our subsequent analysis. As shown in Figure 4, the volatilities of CHO and CHON compounds both concentrate in the range of $-4.5 < \log_{10}(C^*) < 1.5$. In addition, CHO and CHON compounds are randomly distributed in two groups according to O/C and there is no obvious distinction. The

species in the red dashed ellipse are the same as those in the left dashed ellipse in Figure 3, and the compounds in the blue dashed ellipse are the same as those in the right dashed ellipse. The molecular weights of species in the two groups overlapped, although the ones in the red dashed ellipse are characterized with relatively lower molecular weights, and the ones in the blue dashed ellipse are with relatively higher molecular weights. The O/C ratios can clearly distinguish the two groups: $0.56 \pm 0.35$ (average $\pm$ one standard deviation) for the red ellipse group and $0.18 \pm 0.08$ for the blue ellipse group, indicating that the O/C

ratio of these compounds could be a key parameter.

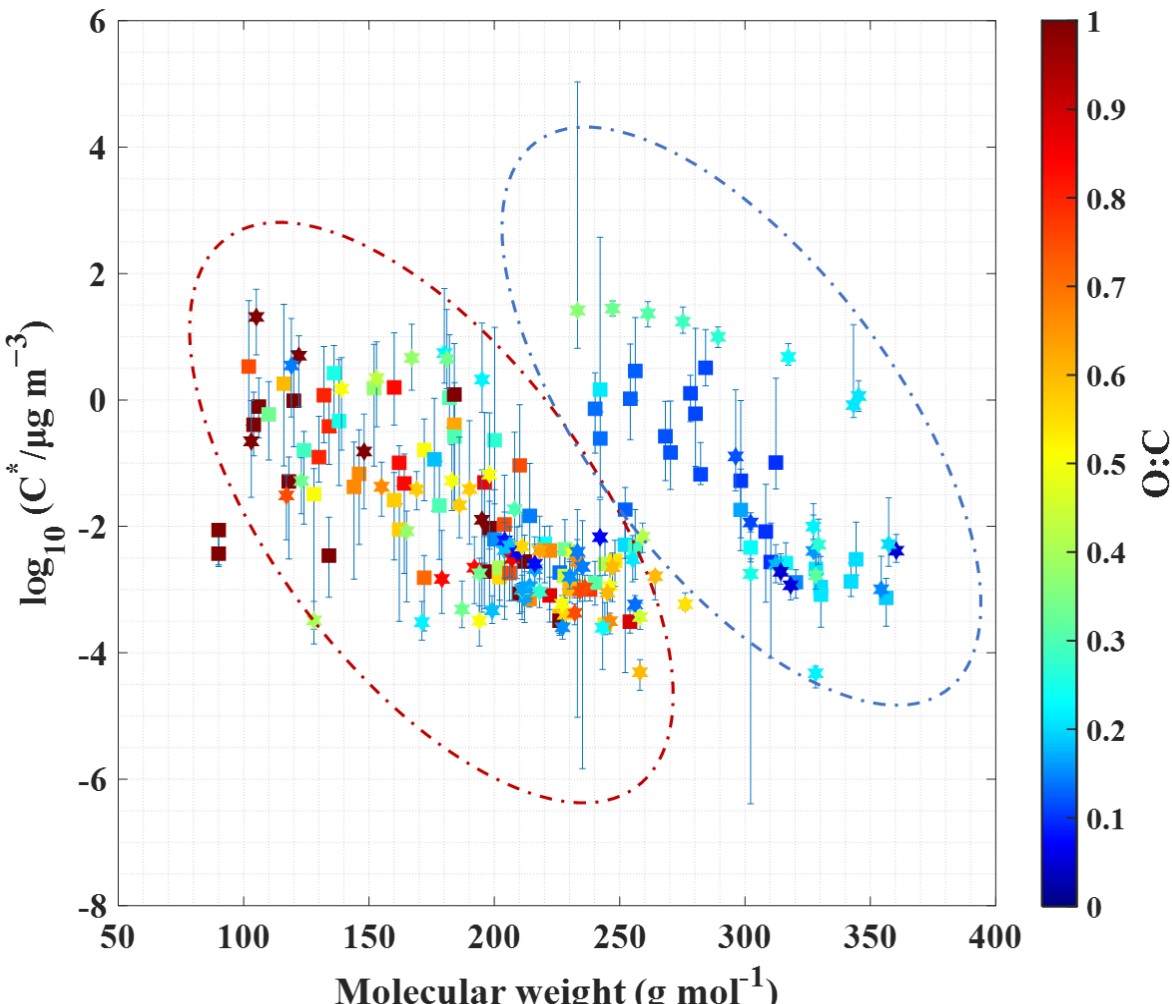

**Figure 4.** Saturation mass concentration of CHO and CHON compounds against their molecular weights, as colour-coded by the O/C ratios. Note that compounds with an O/C ratio equal to or greater than 1.0 are marked with the same colour. The CHO and CHON compounds are denoted by squares and hexagrams, respectively. Whiskers denote 25th and 75th percentile values

of measured saturation mass concentration from 30 ambient samples, and whiskers are ultimately due to variability in the measured $T_{max}$ of CHO and CHON compounds. Dashed ellipses group compounds on the basis of O/C range.

Figure S12 clearly shows the O/C characteristics of compounds in the two regions, where the red triangles correspond to those in the red dashed ellipse of Figure 4, and the blue circles correspond to those in the blue dashed ellipse. The O/C ratios of

organic compounds in the red dashed ellipse ranged from 0.25 to 1.0, and the O/C ratios of those in the blue dashed ellipse

varied between 0-0.25 (Figure S12). $C_6H_{10}O_5$ (levoglucosan or related isomers) in the red dashed ellipse, is a well-accepted tracer of biomass burning OA (BBOA) (Gaston et al., 2016). $C_{16}H_{32}O_2$ (palmitic acid), $C_{17}H_{34}O_2$ (margaric acid), $C_{18}H_{32}O_2$ (linoleic acid), and $C_{18}H_{34}O_2$ (oleic acid) in the blue dashed ellipse, have been previously identified as markers of cooking-influenced OA (COA) (Chow et al., 2007; Pei et al., 2016). The correlation coefficients (*Pearson's r*) between $C_6H_{10}O_5$ and 92% of compounds in the red dashed ellipse are from 0.64 to 0.98, and the correlation coefficients between $C_{18}H_{34}O_2$ and 37% of compounds in the blue dashed ellipse are from 0.60 to 0.74. We show examples of the correlation of $C_6H_{10}O_5$, $C_{18}H_{32}O_2$ and other compounds in Figure S13. Indeed, there are a number of sources of COA and BBOA near the campaign site, and the campaign was carried out during the heating season. Hence, organic compounds in the red dashed ellipse may be mainly derived from BBOA, and those in the blue dashed ellipse may be partly derived from COA.

We thus separately optimized the correlation between the molecular elemental composition and the saturation mass concentration of organic compounds in these two regions in Figure 4. As shown in Table 2, the parameterization of Eq. (4-1) is dedicated to compounds with O/C ratios of 0.25-1, whereas Eq. (4-2) is more suitable for compounds with O/C ratios of 0-0.25. Mohr et al. (2019) derived parameterization mainly based on HOMs ($-11 < \log_{10}(C^*) < 3$) produced by α-pinene oxidation, whereas our fits are mainly based on semi-volatility organic compounds (SVOCs, $10^{-0.5} < C^* \leq 10^{2.5}$) and low-volatility organic compounds (LVOCs, $10^{-4.5} < C^* \leq 10^{-0.5}$), which are predominantly in the particle phase in the atmosphere.

**Table 2.** The volatility parameterizations of this study and cited literature. In this study, the parameterizations of saturation mass concentration were modified by the least-square optimization from Eq. (4) at 298 K.

| | $n_c^0$ | $b_c$ | $b_o$ | $b_{co}$ | $b_N$ | Suggested O/C range | $b_{add}$ |
|---|---|---|---|---|---|---|---|
| Eq. (4-1) in this study | 25 | 0.0700 | 0.6307 | -0.0615 | 2.3962 | 0.25-1 | / |
| Eq. (4-2) in this study | 25 | 0.2075 | 2.8276 | -1.0744 | 1.8223 | 0-0.25 | / |
| Donahue et al. (2011) | 25 | 0.475 | 2.3 | -0.3 | / | / | / |
| Mohr et al. (2019) | 25 | 0.475 | 0.2 | 0.9 | 2.5 | / | / |
| Stolzenburg et al. (2018) (monomers) | 25 | 0.475 | 2.3 | -0.3 | / | / | 0.90 |
| Stolzenburg et al. (2018) (dimers) | 25 | 0.475 | 2.3 | -0.3 | / | / | 1.13 |
| Li et al. (2016) (CHO) | 22.66 | 0.4481 | 1.656 | -0.7790 | / | / | / |
| Li et al. (2016) (CHON) | 24.13 | 0.3667 | 0.7732 | -0.07790 | 1.114 | / | / |

**4 Atmospheric implications**

For ambient studies, it is crucial to develop a more accurate empirical formula to estimate the volatility of organics in particles. Parametrization in Donahue et al. (2011) is mainly based on mono-functional compounds such as alcohol, aldehyde, acid and etc., and could cause a large uncertainty when estimating the volatility of compounds in the range $-5 < \log_{10}(C^*) < 2$ and $1:3 < O/C < 1:1$, because volatilities in this region are extrapolated with volatilities of compounds with simpler molecular formulae (Donahue et al., 2011). In addition, compounds in this region may be characterized with multiple functionalities, which lack reference standards in the Donahue et al. (2011) study. Compounds in this region can be roughly regarded as oxygenated organic aerosols (OOAs) (Donahue et al., 2011). In our study, a yellow dashed frame is used to mark this region in Figure S12, which is occupied by organic compounds that are concentrated in the red dashed ellipse of Figure 4.

We used 15 highly oxygenated organic molecules (HOMs) with O/C ratios of 0.25-1 and 230 CHO compounds with O/C ratios of 0.25-1 as benchmarks to compare the performance of different parameterization methods, as shown in Figure 5a. The volatilities of the 15 HOMs was obtained by SIMPOL from the Tröstl et al. (2016) study. The molecular formulae of 230 CHO (O/C: 0.25-1) compounds are from Zhao et al. (2013) and Mazzoleni et al. (2010), and the molecular structures of these 230

compounds are predicted to be common, i.e., most of function groups of their structures are included in SIMPOL, so that the volatilities of these compounds can be estimated by SIMPOL. As expected, volatilities predicted by the Donahue et al. (2011) parameterization are not completely consistent with those by SIMPOL. Although Mohr et al. (2019) and Stolzenburg et al.

400    (2018) both updated parametrizations based on those 15 HOMs detected by Tröstl et al. (2016), compared to the Mohr et al. (2019) parameterization, the volatility predicted by the Stolzenburg et al. (2018) parameterization does match those by SIMPOL better, which could be attributed to the fact that Mohr et al. (2019) did not separately use parameters for dimer and monomer as Stolzenburg et al. (2018) did, so that the effect of the covalent binding is ignored. On the other hand, the accuracy in predication of volatility of the parameterizations of Stolzenburg et al. (2018), Li et al. (2016), and Eq. (4-1) is generally

405    comparable (Figure 5a). Compared to the parameterizations of Stolzenburg et al. (2018) and Li et al. (2016), the consistency between the parameterization of Eq. (4-1) and SIMPOL is not as good for the more volatile compounds ($\log_{10}(C^*) > 0.5$), but the consistency is better for the LVOCs ($10^{-4.5} < C^* \leq 10^{-0.5}$), which reflects the inherent strength and deficiency of the FIGAERO method, i.e., the FIGAERO method replies on authentic standards that are commonly LVOCs and is thus less suitable for the more volatile compounds. In this study, we used the saturation mass concentration ($C^*$) of five organic standards

410    concentrates in the range of $-5 < \log_{10}(C^*) < 0.5$, as shown in Figure S14. Moreover, the parameterization of Eq. (4-1) was derived based on the ambient compounds in the red dashed ellipse of Figure 4, whose volatilities predominantly concentrate in the range of $-4.5 < \log_{10}(C^*) < 1.5$.

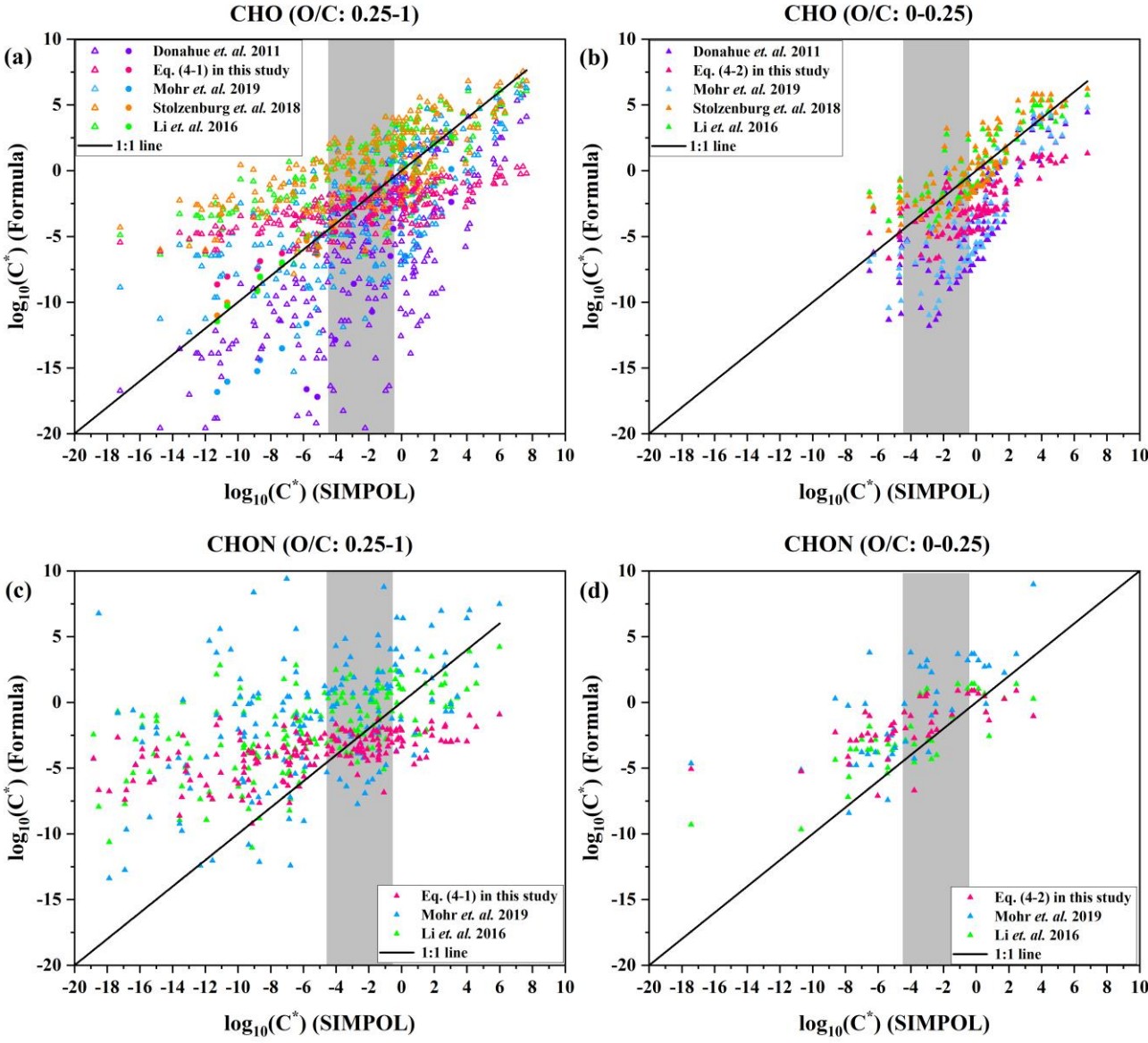

**Figure 5.** Saturation mass concentration ($C^*$) of 15 HOMs (O/C: 0.25-1) and 230 CHO compounds (O/C: 0.25-1) estimated

by Eq. (4-1) **(a)**, 106 CHO compounds (O/C: 0-0.25) estimated by Eq. (4-2) **(b)**, 183 CHON compounds (O/C: 0.25-1) estimated by Eq. (4-1) **(c)**, 46 CHON compounds (O/C: 0-0.25) estimated by Eq. (4-2) **(d)** and the parametrizations from Donahue et al. (2011), Mohr et al. (2019), Stolzenburg et al. (2018), and Li et al. (2016), against that estimated by SIMPOL. In panel a, the 15 HOMs are denoted by circles and 230 CHO compounds are denoted by triangles. The 230 CHO (O/C: 0.25-1), 106 CHO (O/C: 0-0.25), 183 CHON (O/C: 0.25-1) and 46 CHON (O/C: 0-0.25) compounds are from Zhao et al. (2013) and Mazzoleni et al. (2010) field campaigns. The grey colored band denotes low-volatility organic compounds (LVOCs).

Also, we selected 106 CHO compounds with O/C ratios of 0-0.25 from compounds observed by Zhao et al. (2013) and Mazzoleni et al. (2010). The saturation mass concentration ($C^*$) of these compounds are estimated by different parameterizations and SIMPOL. As shown in Figure 5b, compared to parameterizations of Donahue et al. (2011) and Mohr et al. (2019), the parameterizations of Stolzenburg et al. (2018), Li et al. (2016) and Eq. (4-2) are more consistent with SIMPOL. This is mainly because the parameterization of Donahue et al. (2011) was developed according to higher volatility of organic compounds whose volatilities concentrate in the range of $0 < \log_{10}(C^*) < 9$, but those selected 106 CHO compounds with lower volatility whose volatilities concentrate in the range of $-6 < \log_{10}(C^*) < 6$. For the Mohr et al. (2019) parametrization, this could be explained by the fact that those 106 CHO compounds are with lower O/C (0-0.25), but the O/C of 15 HOMs used to update parametrization by Mohr et al. (2019) is 0.25-1. Compared to the parameterizations of Stolzenburg et al. (2018) and Li et al. (2016), in general, the Eq. (4-2) parameterization does not match SIMPOL as well. However, the parameterizations of Stolzenburg et al. (2018), Li et al. (2016) and Eq. (4-2) are comparable for LVOCs. This may be again explained by the difference between the $C^*$ of our organic standards in the literature and those calculated by SIMPOL and by the volatility distribution of organic standards (Figure S14).

We selected 183 CHON compounds with O/C ratios of 0.25-1 and 46 CHON compounds with O/C ratios of 0-0.25 from Zhao et al. (2013) and Mazzoleni et al. (2010). The molecular structures of selected species are assumed to be common. Then their saturation mass concentration ($C^*$) are estimated by different parameterizations and SIMPOL, respectively. The parameterizations of Donahue et al. (2011) and Stolzenburg et al. (2018) relies only on carbon and oxygen numbers and do not explicitly mention the nitrogen coefficient ($b_N$), thus these two parameterizations are excluded from the comparison of the volatility of CHON compounds. As shown in Figure 5c, the performance of Eq. (4-1) parameterization for CHON compounds (O/C: 0.25-1) is similar to that for CHO compounds (O/C: 0.25-1) (Figure 5a). The volatilities LVOCs predicted by the Eq. (4-1) parameterization are more consistent with SIMPOL than the parameterizations of Li et al. (2016) and Mohr et al. (2019). In Figure 5d, the volatilities of CHON compounds (O/C: 0-0.25) predicted by the parameterizations of Li et al. (2016) and our study are comparable, and most of the data points are close to the 1:1 line.

Although the applicability of the parameterizations of Li et al. (2016) and Stolzenburg et al. (2018) is more extensive and the volatilities estimated by these two parameterizations agree well with SIMPOL, Li et al. (2016) and Stolzenburg et al. (2018) modified parameterizations based on a large number of organic species from NCI open database and SIMPOL calculations, respectively. However, the parameterizations of Eq. (4-1) and Eq. (4-2) were derived from organic compounds with different O/C ratios in ambient particles, whose volatilities were estimated by the calibration experiments instead of SIMPOL. Therefore, Eq. (4-1) and Eq. (4-2) can better represent the volatility of ambient organic aerosols. In addition, in Figure 5, for LVOCs, the volatility estimation by Eq. (4-1) and Eq. (4-2) is better than Li et al. (2016) and Stolzenburg et al. (2018). Compared with the ELVOCs, IVOCs and SVOCs, the LVOCs have a dominant contribution to particle growth in new particle formation events. Hence, our parameterizations could be well applied to assess the condensational growth of newly formed particles.

In summary, our study developed empirical volatility-molecular formula functions (Eq. (4-1) and Eq. (4-2)), based on measured $C^*$ of selected CHO and CHON compounds in ambient particles. The parameterizations of Eq. (4-1) and Eq. (4-2) can more accurately predict the volatility of LVOCs with higher O/C (0.25-1) and lower O/C (0-0.25) in the ambient organic aerosols, respectively, owing to the nature of the FIGAERO method. The comparison with previous empirical functions suggests that it is feasible to modify empirical functions based on atmospheric organic compounds with unknown structures and functional

groups using calibration experiments. When analyzing the volatility of atmospheric organic aerosols, it is suggested to create the calibration curve from experiments with the same conditions as that of ambient samples, because the addition of inorganic salts, mixing of organic compounds, mass loadings, and particle size distributions could together influence $T_{max}$ values of

organic compounds. Furthermore, our results suggest that there should be specialized volatility parameterization for different O/C compounds.


*Data availability.* More detailed data can be provided by contacting the corresponding authors.

*Author contributions.* LW designed the study. GY, YLL and YQL conducted the field campaign. SR, YW, YYL, LHW, GY and YLL carried out laboratory experiments. SR analyzed the data. SR, LW and LY wrote the paper with contributions from all of the other co-authors.

*Competing interest.* The authors declare that they have no conflicts of interest.


*Acknowledgement.* This research has been supported by the National Natural Science Foundation of China (21925601, 92044301, 92143301 and 22127811).

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
