# Peer review of "Volatility Parameterization of Ambient Organic Aerosols at a rural site of the Northern China Plain"

_Atmospheric Chemistry and Physics, 2021_

## Author Comment (AC1)

**A point-to-point response to reviewers' comments**

We are very grateful for the helpful and insightful comments from the reviewers, and have carefully revised our manuscript accordingly. In the following point-to-point response, reviewers' comments are repeated in *italics*, whereas our responses are in plain texts labelled with [**Response**]. Line numbers in the responses correspond to those in the revised manuscript (the version with all changes accepted). Modifications to the manuscript are in blue.

**Referee #1**

This work utilized FIGAERO-LToF-CIMS for offline organic aerosol volatility characterization. The authors identified a series of CHO and CHON compounds from ambient samples and developed empirical volatility-molecular formula functions making use of the desorption thermograms that can be obtained by FIGAERO. This study also compared two different methods for laboratory standard compound calibration, which is useful information for FIGAERO users. This paper is overall well written and organized.

**Response:**

We appreciate the positive comments from Reviewer #1 and have revised our manuscript accordingly.

1.One suggestion is that, both CHO and CHON were characterized for ambient samples, but it seems that they were treated the same in subsequent analyses. It would be helpful to label CHO and CHON differently in the figures. In addition, O/C ratio was used to distinguish compounds, which should also be discussed differently for CHO and CHON compounds. More specific comments are described below.

**Response:**

We have labeled CHO and CHON compounds differently by denoting the CHO and CHON compounds with squares and hexagrams, respectively, as shown in Figure R1 (i.e., Figure 4 in the revised manuscript). Also, the CHO and CHON compounds have been discussed separately in the revised manuscript, which (Line 326-328) reads, "As shown in Figure 4, the volatilities of CHO and CHON compounds both concentrate in the range of  $-4.5

**Figure R1.** Saturation mass concentration of CHO and CHON compounds against their molecular weights, as colour-coded by O/C ratios. Note that compounds with an O/C ratio equal to or greater than 1.0 are marked with the same colour. The CHO and CHON compounds are denoted by squares and hexagrams, respectively. Whiskers denote  $25^{\text{th}}$  and  $75^{\text{th}}$  percentile values of measured saturation mass concentration from 30 ambient samples, and whiskers are ultimately due to variability in the measured *Tmax* of CHO and CHON compounds. Dashed ellipses group compounds on the basis of O/C range.

2.Line 14: "Because most standard particulate organic compounds are scarce..." This sentence is incomplete.

**Response:**

This sentence has been revised, which (Line 14-16) reads, "Because most standards for particulate organic compounds are not available, and even for those with standards, their vapor pressures are too low to be measured by most traditional methods.".

3.Line 86: the success of the methods depends on many factors, not only the standards' thermogram characterization. Please clarify here.

**Response:**

We have clarified this point, which (Line 88-89) reads, "Accurately measuring the desorption thermograms of the standards is one of the essential factors for the success of this method.".

4.Line 112: what is the heating temperature ramping rate? The ramping rate can have an influence on the thermal desorption/decomposition process (Yang et al. 2021), and more details here would be helpful.

**Response:**

In our study, the measured heating temperature ramping rate was 2.27 °C/min. Based on the standard of citric acid, Yang et al. (2021) recommends using the faster ramping rate (e.g., 40 °C/min), because the decomposition degree of citric acid under the faster ramping rates of 40 °C/min is lower than that under the slower ramping rate of 3.3 °C/min. Actually, the difference of decomposition degree is only 3.6% between ramping rates of 3.3 °C/min and 40 °C/min. In addition, the fast-ramping rate can result in many decomposition products.

Thus, for ambient organic aerosols, to allow more time to let a larger fraction of molecules evaporate and minimize the formation of decomposition products, the slow ramping rate was adopted.

The detailed description and discussion have been added to the text (Line 118-122), which reads, "The measured ramping rate for heating was 2.27 °C /min in this study. A slower ramping rate allows more time to stay at any momentary desorption temperature so that a larger fraction of molecules would evaporate. Also, a slower ramping rate can separate compounds with similar volatilities better and lead to a smaller number of thermal decomposition products (Lopez-Hilfiker et al., 2014; Yang et al., 2021; Ylisirniö et al., 2021).".

5.Line 138: can the authors explain here how they determine the particle density?

**Response:**

The particle density was determined according to the density of corresponding authentic standards. For example, if there is only one authentic standard in the solution that is used for atomization, the density of atomized and dried particles is adopted as that of this standard. If there are multiple authentic standards in the solution, the particle density will be approximated according to the density and mixing ratio of each standard.

To clearly explain how to determine the particle density, we now state in our revised manuscript (Line 150-151) that "Among them, the particle density can be estimated according to the density of corresponding authentic standards and their mixing ratios in the solution.".

6.Line 149: "It is assumed that the atomized particles were internally mixed with the same mass ratio as that in the solution." Ammonium sulfate is much less volatile than organic compounds mixed with it, and it's highly possible the AS/Org ratio in the particles is higher than that in the solution. More evidence or discussion of the potential bias is needed.

**Response:**

Indeed, our assumption may not well represent the real mixing state of atomized particles. Under our experimental settings, the real AS/Org ratio in the atomized particles cannot be directly measured. The AS/Org ratio could be similar with that in solution and likely have a minor effect on the matrix effect of ammonium sulfate.

We have added the discussion to the revised manuscript, which (Line 164-166) reads, "Although ammonium sulfate is much less volatile than mixed organics and the mixing ratio of ammonium sulfate to organic compounds in atomized particles might be different with that in solution, this assumption likely

**leads to a minor effect on the matrix effect of ammonium sulfate.".**

7. Figure 1: Besides fitted lines, can the author also add the raw data points from each measurement? In addition, can the authors add to the legend the corresponding experiment sets No. (as in Table 1)?

**Response:**

We have added the raw data points and experiment sets No. to the legend of Figure R2 (i.e., Figure 1 in the revised manuscript).

---

## Author Comment (AC3)

**A point-to-point response to reviewers' comments**

We are very grateful for the helpful and insightful comments from the reviewers, and have carefully revised our manuscript accordingly. In the following point-to-point response, reviewers' comments are repeated in *italics*, whereas our responses are in plain texts labelled with **[Response]**. Line numbers in the responses correspond to those in the revised manuscript (the version with all changes accepted). Modifications to the manuscript are in blue.

**Referee #1**

*This work utilized FIGAERO-LToF-CIMS for offline organic aerosol volatility characterization. The authors identified a series of CHO and CHON compounds from ambient samples and developed empirical volatility-molecular formula functions making use of the desorption thermograms that can be obtained by FIGAERO. This study also compared two different methods for laboratory standard compound calibration, which is useful information for FIGAERO users. This paper is overall well written and organized.*

**Response:**

We appreciate the positive comments from Reviewer #1 and have revised our manuscript accordingly.

*1. One suggestion is that, both CHO and CHON were characterized for ambient samples, but it seems that they were treated the same in subsequent analyses. It would be helpful to label CHO and CHON differently in the figures. In addition, O/C ratio was used to distinguish compounds, which should also be discussed differently for CHO and CHON compounds. More specific comments are described below.*

**Response:**
We have labeled CHO and CHON compounds differently by denoting the CHO and CHON compounds with squares and hexagrams, respectively, as shown in Figure R1 (i.e., Figure 4 in the revised manuscript). Also, the CHO and CHON compounds have been discussed separately in the revised manuscript, which (Line 326-328) reads, "As shown in Figure 4, the volatilities of CHO and CHON compounds both concentrate in the range of $-4.5 < \log_{10}(C^*) < 1.5$. In addition, CHO and CHON compounds are randomly distributed in two groups according to O/C and there is no obvious distinction.".

[Figure]

**Figure R1.** Saturation mass concentration of CHO and CHON compounds against their molecular weights, as colour-coded by O/C ratios. Note that compounds with an O/C ratio equal to or greater than 1.0 are marked with the same colour. The CHO and CHON compounds are denoted by squares and hexagrams, respectively. Whiskers denote $25^{th}$ and $75^{th}$ percentile values of measured saturation mass concentration from 30 ambient samples, and whiskers are ultimately due to variability in the measured $T_{max}$ of CHO and CHON compounds. Dashed ellipses group compounds on the basis of O/C range.

*2.Line 14: "Because most standard particulate organic compounds are scarce…" This sentence is incomplete.*

**Response:**
This sentence has been revised, which (Line 14-16) reads, "Because most standards for particulate organic compounds are not available, and even for those with standards, their vapor pressures are too low to be measured by most traditional methods.".

*3.Line 86: the success of the methods depends on many factors, not only the standards' thermogram characterization. Please clarify here.*

**Response:**
We have clarified this point, which (Line 88-89) reads, "Accurately measuring the desorption thermograms of the standards is one of the essential factors for the success of this method.".

*4.Line 112: what is the heating temperature ramping rate? The ramping rate can have an influence on the thermal desorption/decomposition process (Yang et al. 2021), and more details here would be helpful.*

**Response:**

In our study, the measured heating temperature ramping rate was 2.27 °C/min. Based on the standard of citric acid, Yang et al. (2021) recommends using the faster ramping rate (e.g., 40 °C/min), because the decomposition degree of citric acid under the faster ramping rates of 40 °C/min is lower than that under the slower ramping rate of 3.3°C/min. Actually, the difference of decomposition degree is only 3.6% between ramping rates of 3.3 °C/min and 40 °C/min. In addition, the fast-ramping rate can result in many decomposition products.

Thus, for ambient organic aerosols, to allow more time to let a larger fraction of molecules evaporate and minimize the formation of decomposition products, the slow ramping rate was adopted.

The detailed description and discussion have been added to the text (Line 118-122), which reads, "The measured ramping rate for heating was 2.27 ˚C /min in this study. A slower ramping rate allows more time to stay at any momentary desorption temperature so that a larger fraction of molecules would evaporate. Also, a slower ramping rate can separate compounds with similar volatilities better and lead to a smaller number of thermal decomposition products (Lopez-Hilfiker et al., 2014; Yang et al., 2021; Ylisirniö et al., 2021).".

*5.Line 138: can the authors explain here how they determine the particle density?*

**Response:**

The particle density was determined according to the density of corresponding authentic standards. For example, if there is only one authentic standard in the solution that is used for atomization, the density of atomized and dried particles is adopted as that of this standard. If there are multiple authentic standards in the solution, the particle density will be approximated according to the density and mixing ratio of each standard.

To clearly explain how to determine the particle density, we now state in our revised manuscript (Line 150-151) that "Among them, the particle density can be estimated according to the density of corresponding authentic standards and their mixing ratios in the solution.".

*6.Line 149: "It is assumed that the atomized particles were internally mixed with the same mass ratio as that in the solution." Ammonium sulfate is much less volatile than organic compounds mixed with it, and it's highly possible the AS/Org ratio in the particles is higher than that in the solution. More evidence or discussion of the potential bias is needed.*

**Response:**

Indeed, our assumption may not well represent the real mixing state of atomized particles. Under our experimental settings, the real AS/Org ratio in the atomized particles cannot be directly measured. The AS/Org ratio could be similar with that in solution and likely have a minor effect on the matrix effect of ammonium sulfate.

We have added the discussion to the revised manuscript, which (Line 164-166) reads, "Although ammonium sulfate is much less volatile than mixed organics and the mixing ratio of ammonium sulfate to organic compounds in atomized particles might be different with that in solution, this assumption likely

leads to a minor effect on the matrix effect of ammonium sulfate.".

*7.Figure 1: Besides fitted lines, can the author also add the raw data points from each measurement? In addition, can the authors add to the legend the corresponding experiment sets No. (as in Table 1)?*

**Response:**
We have added the raw data points and experiment sets No. to the legend of Figure R2 (i.e., Figure 1 in the revised manuscript).

[Figure]

**Figure R2.** Comparison of calibration results obtained in this study with those reported previously. These solid lines denote the calibration results obtained in this study. Error bars represent ± one standard deviation of $T_{max}$ from four replicate experiments. The light blue dash-dot line denotes calibration results obtained with acids and erythritol by Nah et al. (2019) using the syringe method. The black dash-dot line represents calibration results obtained with 100 ng deposited PEGs (including PEG-4, PEG-5, PEG-6, PEG-7, and PEG-8) by Ylisirniö et al. (2021) using the syringe method. The red dash-dot line denotes calibration results obtained with 100-200 ng PEGs (including PEG-5, PEG-6, PEG-7, and PEG-8) by Ylisirniö et al. (2021) using the atomization method.

*8.Line 235-240: sample NO.4 and NO.5 are different mixtures of different masses, making it hard to compare them. I suggest adding another set of experiments atomizing 500 ng AS + 500 ng Organics (each).*

**Response:**

We have added this set of experiments (i.e., No.5 in Table 1 in the revised manuscript). Furthermore, we have also conducted No.6 and No.8-11 sets of experiments suggested by Reviewer #4 to investigate the influence of mass loadings and other different matrix effects on the thermograms and $T_{max}$. These added experiments have been described in Table R1 (i.e., Table 1 in the revised manuscript), as shown below:

**Table R1.** Conditions of eleven sets of calibration experiments.

| No. | Method | Authentic standards | Concentration | Solvent | Deposited volume | Mass loading |
|---|---|---|---|---|---|---|
| 1 | Syringe deposition | PEG-4 ($C_8H_{18}O_5$)
PEG-5 ($C_{10}H_{22}O_6$)
PEG-6 ($C_{12}H_{26}O_7$)
PEG-7 ($C_{14}H_{30}O_8$)
PEG-8 ($C_{16}H_{34}O_9$) | 0.05 g L$^{-1}$
0.05 g L$^{-1}$
0.05 g L$^{-1}$
0.05 g L$^{-1}$
0.05 g L$^{-1}$ | Acetonitrile | 2 μl | 100 ng
100 ng
100 ng
100 ng
100 ng |
| 2 | Atomization | PEG-4 ($C_8H_{18}O_5$)
PEG-5 ($C_{10}H_{22}O_6$)
PEG-6 ($C_{12}H_{26}O_7$)
PEG-7 ($C_{14}H_{30}O_8$)
PEG-8 ($C_{16}H_{34}O_9$) | 1.0 g L$^{-1}$
1.0 g L$^{-1}$
1.0 g L$^{-1}$
1.0 g L$^{-1}$
1.0 g L$^{-1}$ | Deionized water | / | 500 ng
500 ng
500 ng
500 ng
500 ng |
| 3 | Atomization | Erythritol ($C_4H_{10}O_4$)
PEG-6 ($C_{12}H_{26}O_7$)
PEG-7 ($C_{14}H_{30}O_8$)
PEG-8 ($C_{16}H_{34}O_9$)
Citric acid ($C_6H_8O_7$) | 0.5 g L$^{-1}$
0.5 g L$^{-1}$
0.5 g L$^{-1}$
0.5 g L$^{-1}$
0.5 g L$^{-1}$ | Deionized water | / | 100 ng
100 ng
100 ng
100 ng
100 ng |
| 4 | Atomization | Erythritol($C_4H_{10}O_4$) +ammonium sulfate
PEG-6 ($C_{12}H_{26}O_7$) + ammonium sulfate
PEG-7 ($C_{14}H_{30}O_8$) + ammonium sulfate
PEG-8 ($C_{16}H_{34}O_9$) + ammonium sulfate
Citric acid ($C_6H_8O_7$) + ammonium sulfate | 0.5 g L$^{-1}$ + 0.5 g L$^{-1}$
0.5 g L$^{-1}$ + 0.5 g L$^{-1}$
0.5 g L$^{-1}$ + 0.5 g L$^{-1}$
0.5 g L$^{-1}$ + 0.5 g L$^{-1}$
0.5 g L$^{-1}$ + 0.5 g L$^{-1}$ | Deionized water | / | 200 ng (100 ng + 100 ng)
200 ng (100 ng + 100 ng)
200 ng (100 ng + 100 ng)
200 ng (100 ng + 100 ng)
200 ng (100 ng + 100 ng) |
| 5 | Atomization | Erythritol($C_4H_{10}O_4$) +ammonium sulfate
PEG-6 ($C_{12}H_{26}O_7$) + ammonium sulfate
PEG-7 ($C_{14}H_{30}O_8$) + ammonium sulfate
PEG-8 ($C_{16}H_{34}O_9$) + ammonium sulfate
Citric acid ($C_6H_8O_7$) + ammonium sulfate | 0.5 g L$^{-1}$ + 0.5 g L$^{-1}$
0.5 g L$^{-1}$ + 0.5 g L$^{-1}$
0.5 g L$^{-1}$ + 0.5 g L$^{-1}$
0.5 g L$^{-1}$ + 0.5 g L$^{-1}$
0.5 g L$^{-1}$ + 0.5 g L$^{-1}$ | Deionized water | / | 1000 ng (500 ng + 500 ng)
1000 ng (500 ng + 500 ng)
1000 ng (500 ng + 500 ng)
1000 ng (500 ng + 500 ng)
1000 ng (500 ng + 500 ng) |
| 6 | Atomization | Erythritol($C_4H_{10}O_4$) + PEG-6 ($C_{12}H_{26}O_7$) +PEG-7 ($C_{14}H_{30}O_8$) + PEG-8 ($C_{16}H_{34}O_9$) + citric acid ($C_6H_8O_7$) | 0.5 g L$^{-1}$ + 0.5 g L$^{-1}$ +0.5 g L$^{-1}$ + 0.5 g L$^{-1}$ +0.5 g L$^{-1}$ | Deionized water | / | 1000 ng (200 ng + 200 ng +200 ng +200 ng + 200 ng) |
| 7 | Atomization | Erythritol ($C_4H_{10}O_4$) + PEG-6 ($C_{12}H_{26}O_7$) + PEG-7 ($C_{14}H_{30}O_8$) + PEG-8 ($C_{16}H_{34}O_9$) + citric acid ($C_6H_8O_7$) + ammonium sulfate | 0.5 g L$^{-1}$ + 0.5 g L$^{-1}$ +0.5 g L$^{-1}$ + 0.5 g L$^{-1}$ +0.5 g L$^{-1}$ + 2.5 g L$^{-1}$ | Deionized water | / | 1000 ng (100 ng + 100 ng +100 ng +100 ng + 100 ng + 500 ng) |
| 8 | Atomization | Erythritol ($C_4H_{10}O_4$) + PEG-6 ($C_{12}H_{26}O_7$) + PEG-7 ($C_{14}H_{30}O_8$) + PEG-8 ($C_{16}H_{34}O_9$) + citric acid ($C_6H_8O_7$) + ammonium sulfate | 0.5 g L$^{-1}$ + 0.5 g L$^{-1}$ +0.5 g L$^{-1}$ + 0.5 g L$^{-1}$ +0.5 g L$^{-1}$ + 2.5 g L$^{-1}$ | Deionized water | / | 200 ng (20 ng + 20 ng +20 ng +20 ng + 20 ng + 100 ng) |
| 9 | Atomization | Erythritol ($C_4H_{10}O_4$) + PEG-6 ($C_{12}H_{26}O_7$) + PEG-7 ($C_{14}H_{30}O_8$) + PEG-8 ($C_{16}H_{34}O_9$) + citric acid ($C_6H_8O_7$) + ammonium sulfate | 0.5 g L$^{-1}$ + 0.5 g L$^{-1}$ +0.5 g L$^{-1}$ + 0.5 g L$^{-1}$ +0.5 g L$^{-1}$ + 2.5 g L$^{-1}$ | Deionized water | / | 500 ng (50 ng + 50 ng +50 ng +50 ng + 50 ng + 250 ng) |
| 10 | Atomization | Erythritol ($C_4H_{10}O_4$) + PEG-6 ($C_{12}H_{26}O_7$) + PEG-7 ($C_{14}H_{30}O_8$) + PEG-8 ($C_{16}H_{34}O_9$) + citric acid ($C_6H_8O_7$) + ammonium sulfate | 0.5 g L$^{-1}$ + 0.5 g L$^{-1}$ +0.5 g L$^{-1}$ + 0.5 g L$^{-1}$ +0.5 g L$^{-1}$ + 2.5 g L$^{-1}$ | Deionized water | / | 1500 ng (150 ng + 150 ng +150 ng +150 ng + 150 ng + 750 ng) |

| 11 | Atomization | Erythritol ($C_4H_{10}O_4$) + PEG-6 ($C_{12}H_{26}O_7$) + PEG-7 ($C_{14}H_{30}O_8$) + PEG-8 ($C_{16}H_{34}O_9$) + citric acid ($C_6H_8O_7$) + ammonium sulfate | 0.5 g $L^{-1}$ + 0.5 g $L^{-1}$ +0.5 g $L^{-1}$ + 0.5 g $L^{-1}$ +0.5 g $L^{-1}$ + 2.5 g $L^{-1}$ | Deionized water | / | 2000 ng (200 ng + 200 ng +200 ng +200 ng + 200 ng + 1000 ng) |

According to our new calibration experiments, as shown in Figure R3 (i.e., Figure S5 in the revised Supporting Information (SI)), the mass loadings on Teflon filters, matrix effects of ammonium sulfate, and matrix effects within organic compounds could together affect the $T_{max}$ of organic compounds. Compared with No.4, the $T_{max}$ of organics in No.7 has obviously increased (4 to 21 °C) due to the synergistic effect of the mass loading, matrix effects of ammonium sulfate, and matrix effects within organic compounds (Figure R3). Through comparison of the $T_{max}$ values of No.6 and No.7, a slight $T_{max}$ increase (1 to 4 °C) of five organics was observed owing to the addition of ammonium sulfate. The elevated $T_{max}$ (1 to 8 °C) between No.5 and No.7 were observed due to the matrix effects within organics, which hints the matrix effects within mixed organic compounds could likely be greater than the addition of ammonium sulfate. However, these matrix effects cannot be quantified separately in our study.

We now state in our revised manuscript (Line 254-263) that "Comparison of the $T_{max}$ values of No.6 and No.7 showed that the $T_{max}$ of five organics exhibited a slight increase (1 to 4 °C), which is likely due to the addition of ammonium sulfate. Furthermore, elevated $T_{max}$ (1 to 8 °C) between No.5 and No.7 was observed, which means the matrix effects within mixed organic compounds can also enhance the $T_{max}$ of organics. The $T_{max}$ of five organic compounds increased by 3 to 17 °C with the increase of mass loadings according to No.4 and No.5 (Figure S5). Furthermore, as shown in Figure S6, the $T_{max}$ of five organic compounds generally increases with increased mass loadings, and $T_{max}$ has increased approximately 8 °C as the mass loading increases from 200 ng to 1500 ng. The 95% credible intervals of No.5, No.6 and No.7 experiments are significantly larger than the others, which may be attributed to their higher mass loading (1000 ng) than those in other experiments (100 ng, 200 ng and 500 ng) (Figure S7). Therefore, the $T_{max}$ values of organic compounds are affected together by the addition of ammonium sulfate, the matrix effects within organic compounds, and mass loadings. However, these effects cannot be quantified separately in our study."

To clarify the influence of mass loadings and confidence intervals of fitted lines in Figure 1, Figure S6 and Figure S7 have been added to the revised SI (Line 34-39), which are also shown as Figure R4 and Figure R5 in this response.

[Figure]

**Figure R3.** Influence of addition of ammonium sulfate and mixing of organic compounds on $T_{max}$ of organic standards. The red, blue, light green, magenta and dark green lines denote calibration results of No.3-7 sets of experiments, respectively. Error bars represent ± one standard deviation of $T_{max}$ from four replicate experiments.

[Figure]

**Figure R4.** Influence of mass loading on filters on the $T_{max}$ of organic standards. The grey, dark green and orange lines denote calibration results of No.8, No.7 and No.10 sets of experiments, respectively. Error bars represent ± one standard deviation of $T_{max}$ from four replicate experiments.

[Figure]

**Figure R5.** 95% credible intervals of the fitted lines obtained in this study and previous studies.

*9.Line 285: 181 out of 1448 measured species were included in further analysis. I wonder how much of the total signals can be accounted for by the 181 compounds. Are they the most dominant compounds?*

**Response:**

We now state in our revised manuscript (Line 302-306) that "The 181 species are dominant compounds accounting for 34.1% of the total signal of 1448 compounds. It should be noted that several compounds (e.g., $C_3H_8O_3$, $CH_2O_2$) with high signals in these 1448 compounds were not further analyzed, because they are very volatile at the room temperature (25 ˚C) and their thermograms cannot be characterized with clear $T_{max}$ values from most filter samples.".

*10.Line 306: "The data points for the higher-temperature ones in double-peak thermograms that in fact, do not correspond to a Tmax are removed." How many, if not all, of them are removed?*

**Response:**

78 out of the 181 compounds showed double-peaks in their thermograms, and the $T_{max}$ for the higher-temperature ones in double-peak thermograms of these 78 compounds were removed.

We have added this description to our manuscript, which (Line 325-326) reads, "…, the $T_{max}$ for the higher-temperature ones in double-peak thermograms of 78 compounds are not taken into account.".

**Referee #2**

**General:**

*The authors present the calibrations of Tmax obtained from FIGAERO thermograms using mixed organic and inorganic calibrants, and investigate the effect of ammonium sulfate on the Tmax of several organic standards. Then they use the calibration result from the mixture of ammonium sulfate and five organic standards to derive a formula-based parameterization for the volatility estimation of the organic compounds measured in a rural area in China, and also compare this parameterization with previous parameterizations. Studies on the effect of inorganic species on the thermogram and Tmax behavior of organic compounds are important but scarce. From this point of view, the paper provides new input on this. However, the paper is for now a bit more technical sound, since the scientific discussion or application of this derived parameterization is not enough and feels unfinished. I would therefore recommend that this paper be published on ACP only after major revisions, with more scientific input into the paper.*

**Response:**
We are very grateful for the insightful comments and have revised our manuscript accordingly.

**Major:**

*1.Since the calibration results were the basis for the volatility parameterization for the ambient organic species and substantial discussions on Tmax changes of organic standards were stated to be due to the addition of ammonium sulfate, it would be important to separate the mass loading effects and matrix effects (caused by ammonium sulfate or within these organic standards). For example, Tmax of pure PEG-6 standard increased from 40 degC to ~50 degC with an increasing mass loading from 100 ng to 500 ng (for No2 and No3 experiment set, see Figure S3-S4), which could be due to the mass loading effect. The Tmax of PEG-6 mixed with ammonium sulfate (with a mass loading of 200 ng) didn't change compared to that of pure PEG-6 standard (with a mass loading of 100 ng, for No3 and No4 experiment set, see Figure S4), which may exclude the matrix effect due to the addition of ammonium sulfate.*
*But the Tmax of PEG-6 mixed with ammonium sulfate and other organic standards (with a mass loading of 1000 ng) increased to ~60 degC compared to the Tmax (~50 degC) of PEG-6 mixed with ammonium sulfate (with a mass loading of 200 ng, for No4 and No5 experiment set, see Figure S4). This increase could be due to several reasons such as the matrix effects within these organic standards, higher mass loadings of ammonium sulfate, as well as higher total mass loadings on the filter. Since the mass loading for the calibration experiments varied between 100-1000 ng, discussions on the effect of different mass loadings and potential different matrix effects on the comparison of thermograms and Tmax for this study as well as how this would influence the derived parameterization from ambient observation is necessary.*

**Response:**
We have conducted more laboratory experiments (i.e., No.5-6, and No.8-11 sets of experiments in Table 1 in the revised manuscript) to investigate the influence of mass loadings and different matrix effects on the thermograms and $T_{max}$. Please also refer to our response to Comment #8 from Referee #1 for detailed description.
Owing to these effects on $T_{max}$, calibration experiments must take these matric effects into account,

otherwise the volatilities (C*) of organic compounds would be underestimated. Hence, to accurately evaluate volatilities of ambient aerosols and take these effects into account, the No.7, whose mass loading and the mixing ratio of organics to inorganic are similar to those of ambient filter samples, was selected to derive the calibration curve for ambient organic aerosols.

Selection of the calibration curve to derive parameterization has been described in details in Sect.3.1, which (Line 269-278) reads, "In addition, Ylisirniö et al. (2021) shows that particle size has a moderate impact on the measured $T_{max}$ of organic compounds. The particle size distributions and peak diameters of polydisperse particles in our laboratory experiments (No.4, No.6 and No.7) are similar to those of the ambient samples (Figure S8). Therefore, in our study, particle size distributions have a minor effect on measured $T_{max}$.

To minimize the uncertainties from multiple factors (e.g., the presence of ammonium sulfate, multiple organic compounds, particle size distributions, and mass loading) on $T_{max}$, the calibration line obtained from No.7 was utilized to estimate $T_{max}$ values of organic compounds in ambient particles and to derive our parameterizations, because the experimental conditions of mimic particle samples in No.7 are the closest to those of the ambient samples and can represent ambient organic aerosol particles.".

To clarify the particle size distributions, Figure S8 have been added to the revised SI (Line 40-42), which is also shown as Figure R6 in this response.

[Figure]

**Figure R6.** The particle size distributions of calibration experiments (No.4, No.6 and No.7 sets of experiments) and ambient samples.

*2.The derived parameterization behaves similar as Li et al (2016) and Stolzenburg et al (2018) for the 15 HOMs (O/C:0.25-1), but worse than Li et al (2016) and Stolzenburg et al (2018) for the 132 CHO (O/C:0-*

*0.25). If this is true, I don't see why we should use this parameterization from this study instead of Li et al (2016) or Stolzenburg et al (2018). More scientific discussions on this parameterization method is needed.*

**Response:**

Indeed, the overall quality of our parameterizations is not as good as Li et al (2016) (Figure R7, i.e., Figure 5 in the revised manuscript). However, when compared with SIMPOL, our parameterizations for LVOCs, can well estimate their volatilities, which is better than Li et al (2016) and Stolzenburg et al (2018) (Figure R7). Additionally, our parameterizations were derived from the compounds of ambient organic aerosols, therefore the parameterizations can realistically represent the volatility of ambient organic aerosols. For atmospheric implication, our parameterizations focusing on the estimation of LVOCs can be well applied to the study of new particle formation in ambient air, due to LVOCs dominate the subsequent particle growth after nucleation.

We have added more discussions on our parameterization to the revised manuscript that (Line 427-435) "Although the applicability of the parameterizations of Li et al. (2016) and Stolzenburg et al. (2018) is more extensive and the volatilities estimated by these two parameterizations agree well with the SIMPOL, Li et al. (2016) and Stolzenburg et al. (2018) modified parameterizations based on a large number of organic species from NCI open database and SIMPOL calculations, respectively. However, the parameterizations of Eq. (4-1) and Eq. (4-2) were derived from organic compounds with different O/C ratios in ambient particles, whose volatilities were estimated by the calibration experiments instead of SIMPOL. Therefore, Eq. (4-1) and Eq. (4-2) can better represent the volatility of ambient organic aerosols. In addition, in Figure 5, for LVOCs, the volatility estimation by Eq. (4-1) and Eq. (4-2) is better than Li et al. (2016) and Stolzenburg et al. (2018). Compared with the ELVOCs, IVOCs and SVOCs, the LVOCs have a dominant contribution to particle growth in new particle formation events. Hence, our parameterizations could be well applied to assess the condensational growth of newly formed particles.".

[Figure]

**Figure R7.** Saturation mass concentration ($C^*$) of 15 HOMs (O/C: 0.25-1) and 230 CHO compounds (O/C: 0.25-1) estimated by Eq. (4-1) (**a**), 106 CHO compounds (O/C: 0-0.25) estimated by Eq. (4-2) (**b**), 183 CHON compounds (O/C: 0.25-1) estimated by Eq. (4-1) (**c**), 46 CHON compounds (O/C: 0-0.25) estimated by Eq. (4-2) (**d**) and the parametrizations from Donahue et al. (2011), Mohr et al. (2019), Stolzenburg et al. (2018), and Li et al. (2016), against that estimated by the SIMPOL method. In panel **a**, the 15 HOMs are denoted by circles and 230 CHO compounds are denoted by triangles. The 230 CHO (O/C: 0.25-1), 106 CHO (O/C: 0-0.25), 183 CHON (O/C: 0.25-1) and 46 CHON (O/C: 0-0.25) compounds are from Zhao et al. (2013) and Mazzoleni et al. (2010) field campaigns. The grey colored band denotes low-volatility organic compounds (LVOCs).

**Specific:**

*3. Line 41 – What do the authors mean for the "particle-associated phase"? Is it different from particle phase?*

**Response:**
The particle-associated phase can be regarded as the particle phase. Moreover, according to the suggestion

from Reviewer #4, we focused on describing the methods relevant to CIMS in the section of introduction. Thus, in the revised manuscript, we removed the method relevant to AMS which is based on gas-particle partitioning.

*4.Line 49 –SIMPOL is more a structure-based estimation method of vapor pressure instead of formula-based estimation method. Check a recent work by Isaacmanvanwertz and Aumont, ACP, 2021 (https://acp.copernicus.org/articles/21/6541/2021/). In their work, they also mentioned about Daumit et al (2013) method for formula-based estimation method and modified version of Li et al (2016) method. Please replenish/revise this paragraph.*

**Response:**
We have revised the description about SIMPOL and replenished this paragraph. However, we did not add the Daumit et al. (2013) method, because this paragraph focuses on the molecular formula-based estimation method relevant to CIMS instead of AMS used by Daumit et al. (2013).

We have now stated in our manuscript (Line 46-49) that "The relationship between $C^*$ and molecular formulae of alkane, aldehyde, ketone, alcohol, acid, diol, and diacid was proposed by Donahue et al. (2011), which clarifies the relationship between $n_C$ (the numbers of carbon) and $n_O$ (the numbers of oxygen), and $\log C_0$ . The relationship was derived from a group contribution method SIMPOL that actually is a structure-based estimation method (Pankow and Asher, 2008).",
and (Line 51-55) that "However, Isaacman-Vanwertz and Aumont. (2021) showed that the volatility of CHON compounds estimated by the Li et al. (2016) parameterization is significantly biased by an increase in the number of nitrogen atoms, and thus they modified the nitrogen coefficient for CHON compounds from Li et al. (2016) study by using a fixed relationship between the nitrogen coefficient and the number of the oxygen atom (*i.e.,* $b_N=-2*b_O$).".

*5.Line 112-113 – Please add the flow for the UHP N2 to a 0.1 mCi radioactive Am-241 source, and the ramping rate for the heating.*

**Response:**
The flow rate for the UHP $N_2$ was 1.0 lpm. The ramping rate for heating was 2.27 °C/min.
We have added the flow rate of UHP $N_2$ and ramping rate to our manuscript (Line 115-116) that "In IMR, organic molecules were charged by iodide ions generated by exposure of a 1.0 lpm mixture of $CH_3I$ and UHP $N_2$ to a 0.1 mCi radioactive Am-241 source.",
and (Line 118-119) that "The measured ramping rate for heating was 2.27 ˚C /min in this study.".

*6.Line 141 – It's not very clear whether the authentic organic standards are mixed together within each experiment sets of No. 1-3, or they are injected/atomized one by one? For No. 4 and No.5, it's more clear. Please specify in the texts or Table 1.*

**Response:**
In No.1-3 sets of experiments, the authentic organic standards were atomized one by one, not mixed together. The No.1-3 have been described in the same way as No.4 in Table R1 (i.e., Table 1 in the revised manuscript).

*7.Line 156 – For Table 1, the mass loading for different experiments varies. According to Huang et al*

*(2018; https://acp.copernicus.org/articles/18/2883/2018/) and Wang and Ruiz (2018; https://acp.copernicus.org/articles/18/15535/2018/), different mass loading on the filter could influence the thermogram shape and Tmax of organic compounds. It seems the author also observed similar behavior, if one compare the Tmax of PEG-6,7,8 from No.2 and No.3 experiment sets in Figure S3 and S4. Since both No.2 and No.3 experiment sets are done by atomization method but with different mass loadings on the filter, Tmax values are found to differ. For example, with 100 ng of mass loading Tmax of PEG-6 was 40 degC (Figure S4), but with 500 ng of mass loading its Tmax increased to ~50 degC (Figure S3).*

*Since the study is based on the Tmax calibration, could the authors comment on the effect of different mass loadings on the comparison of thermograms and Tmax for this study as well as how this would influence the derived parameterization from ambient observation?*

**Response:**

Please refer to our response to Comment #8 from Referee #1 and to Comment #1 from Referee #2.

*8.Line 163-166 – What's the aerosol mass loading on the filter? Please add this information.*

**Response:**

We have added the information on aerosol mass loading to the revised manuscript, which (Line 190-192) reads, "In this study, 30 filter samples between January 15, 2019 and January 22, 2019 were analyzed with FIGAERO offline, because mass loadings of these 30 filter samples varied from 200 ng to 3500 ng with a median of 1100 ng, which is similar to those in the calibration experiments.".

*9.Line 170-190 – Could the authors comment on the uncertainty for the C\* calculation as well as the uncertainty using this parameterization method?*

**Response:**

The uncertainty for the C\* calculation was affected by many factors, such as the calibration methods (i.e., syringe and atomization), ramping rates, particle size distributions, and mass loading on filters. However, the contribution of each factor to the total uncertainty of C\* calculation cannot be quantified.

The calibration curve method was first proposed by Bannan et al. (2019), they selected a variety of carboxylic acid species as benchmarks to compare literature values (*P*sat1) of vapour pressure with calculated vapour pressures (Psat2) using the PEG calibration curve. There is a good agreement between the FIGAERO (Psat2) and literature vapour pressures (Psat1). Δ [$\log_{10}$(Psat)] of carboxylic acids ranged from 0.001 to 0.824, and the total uncertainty of Eq. (1), is estimated to be less than 26%.

The overall uncertainty of parameterization method may be calculated by the uncertainty of each parameter, i.e., $b_c$, $b_o$, $b_N$ and $b_{co}$. However, these parameters are obtained mainly based on empirical fitting. Therefore, it is difficult to estimate the uncertainty of each parameter.

*10.Line 226-229 – From Figure S4 caption, it seems the No.3 set was done for each standard one by one using atomization method. If that is the case, the figure shows the PEGs 6-8 $T_{max}$ for No.4 set (AS+each standard) was similar to those for No. 3 set (Each standard). It thus would indicate the increase of PEGs 6-8 $T_{max}$ for No. 5 set (AS+five standards) is probably not (only) due to the matrix effects caused by the addition of ammonium sulfate, but due to the matrix effects within these organic standards. Besides, as*

*the authors mentioned in Line 237-240, the $T_{max}$ increase could also be due to higher mass loadings for No. 5 set. It would be important to separate these different reasons, i.e., matrix effects within organic standards or due to the addition of ammonium sulfate, or mass loading effects.*

**Response:**
We have conducted more laboratory experiments (No.5-6, 8-11 sets of experiments in the revised Table 1) to separate these different effects.
Please refer to our response to Comment #8 from Referee #1.

*11.Line 229-231 – Clear connection between more partitioning of organic acids and lower volatility of particulate organic compounds is missing. For example, partitioning of SVOC would increase the volatility of particles. Please explain/clarify a bit more.*

**Response:**
According to the question of referee #4, the particles were dried immediately after the nebulizer and then collected in a short time. It is unlikely to form organic salt between organics and ammonium sulfate to facilitate partitioning of organic acid. Therefore, this sentence has been modified in the revised manuscript.

*12.Line 245-247 – As for the calibration line is similar to that of Nah et al (2019), do the authors mean the lines are quite close to each other? But the slope is a bit different from that of Ylisirniö et al (2021) and Nah et al (2019). Could it be due to Nah et al (2019) used acids with O/C <0.25?*

**Response:**
We mean these two lines are quite close to each other. We have used "quite close" instead of "similar" in the revised manuscript. The slope of the calibration line in this study is a bit different from that of Nah et al. (2019), which could be due to O/C differences of organic standards, i.e., Nah et al (2019) used acids with O/C < 0.25.

We have now stated in the revised manuscript (Line 240-243) that "The solid calibration line obtained with PEGs (O/C > 0.25 ) by the syringe deposition method in this study is quite close to the dash-dot calibration line that was also obtained by the syringe deposition method with acids (O/C < 0.25 and O/C > 0.25 ) and erythritol (O/C > 0.25 ) (Nah et al., 2019). Yet the slopes of two lines are a bit different, which may be attributed to different O/C ratio of organic standards.".

*13.Line 259-261 – It would be informative to add the fitted equation (or the fitted parameters a and b of Eq. (3)) of No.5 experiment set here or in Figure 1.*

**Response:**
We have added the fitted parameters to Figure R2 (i.e., Figure 1 in the revised manuscript).
Please refer to our response to Comment #7 from Referee #1.

*14.Line 264-267 – Please add the contribution of each group (CHO, CHON, other, unidentified) to the total signal in brackets after each group.*

**Response:**
We have added the contribution of each group in the revised manuscript (Line 281-285), which reads,
"Among them, 340 CHO and 663 CHON species account for 43.5% and 20.8% of the total signals,

respectively, because the iodide-adduct chemical ionization is sensitive toward multifunctional oxygenated organic compounds with minimal fragmentation (Bertram et al., 2011; Lopez-Hilfiker et al., 2016). In addition to 326 other species (30.8% of the total signals) that have been assigned with molecular formulae but cannot be divided into either CHO or CHON groups, there are 119 species (4.9% of the total signals) without attributed molecular formulae.".

*15. Line 280-281 – What are these dominated compounds and their potential sources? C6H10O5 could be levoglucosan from biomass burning. How about the others?*

**Response:**
The dominated compounds contain $C_{13}H_{25}NO_2$, $C_{16}H_{32}O_2$, $C_{18}H_{35}NO_4$, $C_6H_{10}O_5$, $C_9H_{17}NO_2$, and $C_{18}H_{34}O_2$. The compounds $C_{16}H_{32}O_2$ and $C_{18}H_{34}O_2$ could be palmitic acid and oleic acid, respectively. These two compounds are often utilized as tracers of COA, so that their potential sources could be cooking. The potential sources of $C_6H_{10}O_5$, $C_{16}H_{32}O_2$ and $C_{18}H_{34}O_2$ have been described (Line 345-348) in the revised manuscript.

$C_{13}H_{25}NO_2$, $C_{18}H_{35}NO_4$, and $C_9H_{17}NO_2$ are not reported in previous studies. Correlation coefficients (Pearson's r) between $C_{18}H_{34}O_2$, $C_6H_{10}O_5$ and these three compounds are all less than 0.6. Therefore, we do not attempt to identify potential sources for these three compounds.

*16. Line 324-330 – Would be nice to mark C6H10O5, C16H32O2, C17H34O2, C18H32O2, and C18H34O2 in Figure S7.*

**Response:**
$C_6H_{10}O_5$, $C_{16}H_{32}O_2$, $C_{17}H_{34}O_2$, $C_{18}H_{32}O_2$, and $C_{18}H_{34}O_2$ have been marked in Figure R8 (i.e., Figure S11 in the revised SI).

[Figure]

**Figure R8.** The volatility ($\log_{10}(C^*)$) for 90 CHO and 91 CHNO species versus O/C ratios. Red triangles and blue circles denote the compounds in the red and blue dashed ellipses in Figure 4, respectively.

*17.Line 338-341 – The authors mentioned in Line 280-281 about the dominated compounds including C13H25NO2, C9H17NO2 etc. It seems these CHON with nO<=2 are quite important. Would the Equation (4) still be applicable to them since the equation subtract 3nN for each nO? Besides, the Equation 4 is a modified parameterization in Mohr et al (2019) specified for HOM with big nO. Maybe the modified Li et al (2016) parameterization equation by Isaacmanvanwertz and Aumont, ACP, 2021 would be a better option considering the fits in Table 2 are mainly based on SVOC and LVOC?*

**Response:**
Actually, in ambient aerosols, the number of CHON species with nO≥3 (63 out of 90 CHON compounds in this study) is much larger than that of CHON with nO≤ 2 (27 out of 90 CHON compounds in this study). Thus, we modified Eq. (4) (equation of Mohr et al (2019)) that subtracts 3nN for each nO in order to achieve a better estimation on the volatility of ambient organic aerosols.
Furthermore, Eq. (4-1) was obtained using organic compounds including 22 CHON with nO≤2 and 47 CHON with nO≥3, and Eq. (4-2) was obtained using organic compounds including 5 CHON with nO≤2 and 16 CHON with nO≥3. In Eq. (4-1) and Eq. (4-2), CHON species were both used to derive our parameterizations. Therefore, the parameterizations in this study could be applicable to ambient species with small nO (e.g., CHON with nO≤2).

We have tried to fit the parameterization using the equation of Isaacmanvanwertz and Aumont. (2021) based on 181 compounds in this study. We also used a number of CHO and CHON compounds that are different from these 181 compounds and contain SVOCs and LVOCs to compare the performance of this parameterization with our parameterization, as shown in Figure R10. The volatilities of CHO and CHON predicted by these two parameterizations are very similar. Therefore, the modified Li et al. (2016) parameterization equation by Isaacmanvanwertz and Aumont. (2021) is not significantly better than the equation of Mohr et al. (2019) for SVOCs and LVOCs.

[Figure]

**Figure R9.** Saturation mass concentrations ($C^*$) of CHO and CHON estimated by the parameterizations

in this study against those estimated by fitted parameterizations based on the equation of Isaacmanvanwertz and Aumont. (2021).

*18.Line 377-379 – Since Isaacman-vanwertz and Aumont (2021) found that the vapor pressures of CHON compounds estimated by Li et al. (2016) significantly biased with an increase of the number of nitrogen atoms, it would be important to add the comparison of the logC\* (Formula) vs. logC\* (SIMPOL) for CHON compounds as an additional panel in Figure 5.*

**Response:**
We have added the comparison of logC\* (Formula) versus logC\* (SIMPOL) for CHON (O/C: 0.25-1) and CHON (O/C: 0-0.25) compounds as shown in Figure R7 (c) and Figure R7 (d) (i.e., Figure 5 (c) and Figure 5 (d) in the revised manuscript).

The discussion of the intercomparison between logC\* (Formula) and logC\* (SIMPOL) has been added to our revised manuscript (Line 415-425) that "We selected 183 CHON compounds with O/C ratios of 0.25-1 and 46 CHON compounds with O/C ratios of 0-0.25 from Zhao et al. (2013) and Mazzoleni et al. (2010). The molecular structures of selected species can be reliably predicted, and then their saturation mass concentration ($C^*$) are estimated by different parameterizations and SIMPOL, respectively. The parameterizations of Donahue et al. (2011) and Stolzenburg et al. (2018) relies only on carbon and oxygen number and do not explicitly mention the nitrogen coefficient ($b_N$), thus these two parameterizations are excluded from the comparison of the volatility of CHON compounds. As shown in Figure 5c, the performance of Eq. (4-1) parameterization for CHON compounds (O/C: 0.25-1) is similar to that for CHO compounds (O/C: 0.25-1) (Figure 5a). The volatilities LVOCs predicted by the Eq. (4-1) parameterization are more consistent with SIMPOL than the parameterizations of Li et al. (2016). In Figure 5d, the volatilities of CHON compounds (O/C: 0-0.25) predicted by the parameterizations of Li et al. (2016) and our study are comparable, and most of the data points are close to the 1:1 line. The volatility of CHON compounds predicted by the Mohr et al. (2019) parameterization does not match those by SIMPOL very well.".

*19.Line 397-404 – In general Eq. (4-1) and (4-2) parameterization behaves better than Donahue et al (2011) and Mohr et al (2019). Based on Figure 5, Eq. (4-1) behaves similar as Li et al (2016) and Stolzenburg et al (2018) for the 15 HOMs (O/C:0.25-1), but Eq. (4-2) behaves worse than Li et al (2016) and Stolzenburg et al (2018) for the 132 CHO (O/C:0-0.25). If this is the case, could the authors comment on why we should use this parameterization from this study instead of Li et al (2016) or Stolzenburg et al (2018)?*

**Response:**
Please refer to our response to Comment #2 from Referee #2.

**Technical:**

*20.Line 17 – Change to "formulae" throughout the manuscript.*

**Response:**
We have replaced "formulas" with "formulae" throughout the manuscript.

*21.Line 30 – Change to "for organic compounds with different O/C ratios".*

**Response:**
We have revised our manuscript accordingly.

*22.Line 33 – Either say "a significant mass fraction" or "total submicron particulate mass".*

**Response:**
We have revised our manuscript accordingly.

*23.Line 110 – A typo for "UHP N2".*

**Response:**
This typo has been revised.

*24.Line 197 – Refer to Figure S3 when describing the results.*

**Response:**
We have revised our manuscript accordingly.

*25.Line 224 and 226 – Refer to Figure S4 when describing the results.*

**Response:**
We have revised our manuscript accordingly.

*26.Line 350 – Either "O:C" or "O/C". Make it consistent throughout the text.*

**Response:**
We have replaced "O:C" with "O/C" throughout the manuscript.

*27.Line 399-404 – Too long sentence. Please reformulate.*

**Response:**
We have reformulated our manuscript accordingly.

**Referee #3**

*In the manuscript "Volatility Parameterization of Ambient Organic Aerosols at a rural site of the Northern China Plain", Ren et al. report on the results of thermal desorption mass spectrometry measurements of the organic aerosol component of filter-collected ambient aerosol. The work also features a convincing effort in carefully "calibrating" the desorption method so that effective volatilities (or vapor pressures, saturation concentrations C*) can be inferred for individual organic compositions. The calibration experiments did not turn out as good as they maybe could have, and much of the ambient data (section 3.2) at least appears to have been discarded in favor of focusing on easier-to-analyze signals. An additional data selection criteria, however, was the continued prevalence of the respective compositions, which makes the selection particularly useful despite its potential narrowness. Importantly, the calibrations allow a thorough analysis of the observations. The authors explore how the resulting C\* values could be parameterized based on compositions. They compare their findings to previous attempts in the literature, but which have to a large part been relying on calculations using group contribution theory to extrapolate to compositions observed in organic aerosol. Thereby, this study's analysis of thermal desorption mass spec data goes deeper than what is often provided by other studies using similar datasets. For these reasons, I believe the manuscript is of considerable interest and deserves publication in Atmospheric Chemistry & Physics, although I do suggest substantial "polishing" before that, considering my comments below. I also stumbled upon an apparent contradiction in the analysis, e.g. when comparing Figs. 3-4 with Fig. 5. I am elaborating on that in my comments as well. Some clarification is at least warranted, or possibly some semi-major revision.*

**Response:**
We appreciate the insightful and positive comments from Reviewer #3 and have revised our manuscript accordingly.

**General notes:**

*1. I noted that only one week's worth of ambient was being used in this study. Was there particular reasons for that?*

**Response:**
Actually, we conducted a 35-day field campaign in winter in the Beijing-Tianjin-Hebei region. Particle pollution was frequent and severe in winter in this region. During our 35-day field campaign, there were 25 days with $PM_{2.5}$ concentrations above 150 μg/m³, which led to very high mass loadings (from 150 to 15000ng) on collected filters. High mass loadings can cause reagent ion depletion when the filter samples were measured by CIMS resulting in the inaccurate quantification of organics. Besides, high mass loadings also have a significant effect on $T_{max}$ values. Thus, to avoid the effects of high mass loading on CIMS measurement and $T_{max,}$ we selected only 30 filter samples from 8 days, with moderate mass loadings from 200 to 3500 ng with a median of 1100 ng, which is similar to that of our calibration experiment (No.7 in Table R1) to be further analyzed by FIGAERO -CIMS.

We have added the information on mass loadings to our manuscript, which (Line 190-192) reads, "In this study, 30 filter samples between January 15, 2019 and January 22, 2019 were analyzed with FIGAERO

offline, because mass loadings of these 30 filter samples varied from 200 ng to 3500 ng with a median of 1100 ng, which is similar to those in the calibration experiments.".

*2.An additional result that might be worth looking into, similar to the analysis using retrieved $T_{max}$ values, would be the **widths** of the fitted peaks, which I understand were largely unconstrained.*
*For example, could those widths tend to change with increasing/decreasing molecular weights, O/C or Tmax? And most importantly: could they help identifying decomposition processes?*

**Response:**
We used the Gaussian function to fit the thermogram results of calibration experiments. The widths of the fitted peaks, molecular weights, O/C ratios, and $T_{max}$ of five standards in the No.7 experiments have been summarized in Table R2. The correlation coefficients (*Pearson's r*) between widths of the fitted peaks and molecular weights, O/C and $T_{max}$ are shown in Table R3. No significant correlations were observed between widths of the fitted peaks and molecular weights and O/C ratios. Yet, it shows a good anti-correlation between widths of the fitted peaks and $T_{max,}$ which could be attributed to that our heating temperature ramping rate was slow and more time was needed for organic compounds with lower volatilities (i.e., higher $T_{max}$ values) to volatilize during thermal desorption, resulting in wider peaks.

In general, the second peaks (higher temperature peaks of the thermograms), can be likely attributed to the decomposition of lower volatility compounds (Lopez-Hilfiker et al., 2015). Thus, the decomposition processes were mainly identified based on the independent second warmer peaks. The widths of the fitted peaks may not help with identification of decomposition processes.

Table R2. The averages of widths of the fitted peaks, molecular weights, O/C and $T_{max}$ of five organic standards in four replicate experiments.

| | Widths (Th) | Molecular weights (Da) | O/C | $T_{max}$ (°C) |
|---|---|---|---|---|
| Erythritol ($C_4H_{10}O_4$) | 17.34 | 122.12 | 1.00 | 49.09 |
| PEG6 ($C_{12}H_{26}O_7$) | 17.82 | 282.33 | 0.58 | 58.79 |
| PEG7 ($C_{14}H_{30}O_8$) | 15.14 | 326.39 | 0.57 | 81.43 |
| PEG8 ($C_{16}H_{34}O_9$) | 13.56 | 370.44 | 0.56 | 90.73 |
| Citric acid ($C_6H_8O_7$) | 9.50 | 192.13 | 1.17 | 100.94 |

Table R3. The correlation coefficients (*Pearson's r*) between widths of the fitted peaks and molecular weights, O/C and $T_{max}$.

| | Widths |
|---|---|
| Molecular weights (Da) | 0.112 |
| O/C | -0.491 |
| $T_{max}$ (°C) | -0.919* |
| * $p < 0.05$ | |

**Technical comments:**

*3.Check that peer-reviewed papers are being cited rather than their discussions papers (e.g. ACPD and AMTD), where available.*

**Response:**

We have updated the cited papers, which (Line 201) reads, "On the other hand, $P_{sat}$ can be converted to $C^*$ with the assumption of the ideal gas law (Ylisirniö et al., **2020**, 2021).".

*4.There are many (individually minor) grammatical and semantical mistakes here and there. The manuscript is readable overall, but they do impede comprehensive reading at some places. Some of the semantical mistakes at least will also confuse readers, in particular if not familiar with the used methodology. I suggest appropriate proof-reading/language checks.*

**Response:**
Thank you for pointing this out. We have proofread our manuscript and corrected a number of grammatical and semantical mistakes.

**Major specific comments:**

*5.Lines 367-369: I am not following here. Aren't the compounds used for the "Eq. (4-1)" parametrization by definition containing (exclusively) OOA species?*

**Response:**
Indeed, the compounds used for the Eq. (4-1) parametrization contain OOA species. Since we have added more CHO compounds (O/C: 0.25-1) to Figure 5a, this part has been rewritten. Please refer to the response to Comment #6 of Referee #3.

*6.In any case, however, the fair agreement with SIMPOL predictions is remarkable (even though only shown for 15 selected compounds) and worth pointing it, as it follows directly from calibrations and measurements, without using SIMPOL calculations. Whereas, if I remember correctly, the other cited works (Li, Tröstl, Stolzenburg) indeed based their parametrizations on SIMPOL calculations (so their agreement is expected).*
*Beyond Fig. 5a, however, it would be interesting to see how the authors' parametrization, i.e., Eq. (4-1), worked out for the other compounds in this group? I.e., do measurement derived C\* agree with SIMPOL also for other compositions in the high-O/C group, besides the 15 examples shown? That would however require making assumptions on molecular structures and new SIMPOL calculations. But the authors did that for members of the low- O/C group (Fig. 5b), so my suggestion might be relatively straight forward to implement. (But not sure which compounds were used in Fig. 5b see comment below also ("Moreover,....").)*

**Response:**
More CHO compounds (O/C: 0.25-1) have been added to intercompare with other parametrizations in Figure R7(a) (i.e., Figure 5a in revised manuscript).

We now state in our revised manuscript (Line 376-380) that "Furthermore, we used 15 highly oxygenated organic molecules (HOMs) with O/C ratios of 0.25-1 and 230 CHO compounds with O/C ratios of 0.25-1 as benchmarks to compare the performance of different parameterization methods, as shown in Figure 5a. The volatilities of 15 HOMs was obtained by SIMPOL from the Tröstl et al. (2016) study. The 230 CHO (O/C: 0.25-1) compounds are from Zhao et al. (2013) and Mazzoleni et al. (2010), because predicted molecular structures of these 230 compounds are reliable. ",

and (Line 384-391) that "On the other hand, the accuracy in predication of volatility of the parameterizations of Stolzenburg et al. (2018), Li et al. (2016), and Eq. (4-1) is generally comparable (Figure 5a). Compared to the parameterizations of Stolzenburg et al. (2018) and Li et al. (2016), the consistency between the parameterization of Eq. (4-1) and SIMPOL is not as good for the more volatile compounds ($\log_{10}(C^*) > 0.5$), but the consistency is better for the LVOCs ($10^{-4.5} < C^* \leq 10^{-0.5}$), which reflects the inherent strength and deficiency of the FIGAERO method. One of reasons may be that in this study, we used the saturation mass concentration ($C^*$) of five organic standards concentrates in the range of $-5 < \log_{10}(C^*) < 0.5$, as shown in Figure S13. Moreover, the parameterization of Eq. (4-1) was derived based on the ambient compounds in the red dashed ellipse of Figure 4, whose volatilities predominantly concentrate in the range of $-4.5 < \log_{10}(C^*) < 1.5$.".

Figure S13 has been added to the revised SI (Line 61-63) which is also showed as Figure R10 in the response.

[Figure]

**Figure R10.** Saturation mass concentration (C*) of five organic compounds calculated by SIMPOL against that from Krieger et al. (2018), Emanuelsson et al. (2016), and Ye et al. (2019).

*7.Lines 391-399: I am lost a bit again. I had to read the first sentence (lines 390-393) several times, as I failed to understand for a while which 42 compounds were being referred to.*

**Response:**
The 42 compounds are alcohols, aldehydes, acids, and diols with O/C ratios of 0-0.25, and selected from NIST. These 42 compounds are not relevant to the CHO (O/C: 0-0.25) compounds in Figure 5(b) from the Zhao et al. (2013) and Mazzoleni et al. (2010) field campaigns. Since the new Figure 5 in the revised manuscript has been able to represent the application of our parametrizations, Figure S9 and descriptions associated with these 42 compounds have been removed.

*8.More generally, the paragraph here spends a lot of time arguing why the "Eq. (4-2)" parametrization agrees poorly with NIST data, whereas SIMPOL-based parametrizations perform better (Fig. S9). The point raised about this study's parametrization being based on observations of compounds with generally lower volatility (at least as inferred by Tmax) may be part of the reason. But the main point, which may be lost, is that the agreement with SIMPOL is also poor for the other 132 (=observed?) compounds (Fig. 5b). The questions I would then have are: (1) is there a reason for SIMPOL to be less accurate for compounds with lower O/C? or (2) is there a reason for the "Eq. 4-2" parametrization to be less accurate, i.e., for the inference of C\* based on Tmax to be not or less valid? In other words, the results shown in Fig. S9, although interesting, may distract/confuse the more important points of discussion in the main text.*

**Response:**

We used 42 compounds (O/C: 0-0.25) from Figure S9 in the original submission as benchmarks to compare the results estimated by SIMPOL with experimental data from NIST. In Figure R11, the volatilities of these 42 compounds with low O/C predicted by SIMPOL agree with values in NIST. Therefore, it is unlikely that SIMPOL is less accurate for compounds with low O/C.

The FIGAERO method is based on the correlation between the $T_{max}$ and the vaporization enthalpy of organic compounds to estimate the volatility (C\*) (Lopez-Hilfiker et al., 2014). The reliability of FIGAERO method is almost consistent for organic compounds with various O/C ratios. We fitted parameterizations of Eq. (4-1) and Eq. (4-2) both using the FIGAERO method, so the accuracy of Eq. (4-1) and Eq. (4-2) should be the same.

Please refer to the response to Comment #6 of Referee #3.

To clarify the reasons, we now state in our revised manuscript (Line 409-413) that "Compared to the parameterizations of Stolzenburg et al. (2018) and Li et al. (2016), in general, the Eq. (4-2) parameterization does not match SIMPOL as well. However, the parameterizations of Stolzenburg et al. (2018), Li et al. (2016) and Eq. (4-2) are comparable for LVOCs. This may be again explained by the difference between the C\* of our organic standards in the literature and those calculated by SIMPOL and by the volatility distribution of organic standards (Figure S13).".

[Figure]

**Figure R11.** Saturation mass concentration (C*) of 42 alcohols, aldehydes, acids and diols with O/C from 0 to 0.25 estimated by SIMPOL, against that from NIST.

*9.Moreover, it remains unclear which those 132 selected compounds for Fig. 5b are. Is there substantial overlap with observed compositions, or possibly even hardly any?*
*And a key message of Fig. 5b is that parametrization Eq4-2 gives TOO LOW C\* values (compared to SIMPOL). On the other hand, Figs. 3 and 4 suggest that parametrization Eq4-2 would yield overall HIGHER C\* values than parametrization Eq4-1 (which agrees well with SIMPOL, Fig. 5a).*
*What am I missing?*

**Response:**
26 out of 132 compounds are from this study. For a more objective comparison, we have removed 26 compounds that overlap with the observed 181 compounds in this study, and the remaining 106 compounds are all from Zhao et al. (2013) and Mazzoleni et al. (2010) field campaigns in the revised Figure 5b.
To clarify the selection of compounds, we now state in our revised manuscript (Line 401-402) that "Also, we selected 106 CHO compounds with O/C ratios of 0-0.25 from compounds observed by Zhao et al. (2013) and Mazzoleni et al. (2010). ".

The Eq. (4-1) parametrization and Eq. (4-2) parametrization are dedicated to compounds with O/C ratios of 0.25-1 and 0-0.25, respectively. The O/C ratio is an important parameter that determines whether Eq. (4-1) or Eq. (4-2) is used to estimate the C* values of compounds. The volatility of an organic compound can only be estimated by one of equations on the basis of O/C ratios. Therefore, the Eq. (4-1) parametrization and Eq. (4-2) parametrization cannot be directly compared.

*10.As a consequence, I would be more careful in the final summary of atmospheric implications (lines 410+). I agree that Eq. 4-1 is doing quite well (for higher-O/C lower- MW compounds), but I am not*

*convinced about Eq. 4-2 (for lower-O/C higher-MW compounds). At least I would not go as far as to claim that it is "more accurate" (and more than what?) for ambient aerosol. The authors hypothesize that interactions with inorganic aerosol components play a more important role for low-O/C compounds, thus lowering their effective C\*. That hypothesis is plausible in principle, and I agree that those interactions are insufficiently studied. But here, it remains rather poorly supported by some discrepancies during calibration. (In the minor comments below, I also suggest an alternative hypothesis for those discrepancies.) Alternative hypotheses would be warranted too. For example, could those higher-MW compounds be structurally different in some fundamentally difference for SIMPOL to stop working? Is there a possibility for the FIGAERO method to be less reliable for those compounds?*

**Response:**

We have added CHON compounds (O/C: 0-0.25) to intercompare with other parametrizations, the volatilities of CHON compounds estimated by Eq. (4-2) are close to that estimated by the SIMPOL method. Therefore, the original hypothesis that interactions with inorganic aerosol components play a more important role in low-O/C compounds maybe infeasible and has been corrected.

Also, we have analyzed the proposed two alternative hypotheses. Please refer to our response to Comment #8 from Referee #3.

*11. And besides that, I still don't quite understand the discrepancy between the following (see also comment above, "Moreover,..."):*
*- Eq. 4-2 gives LOWER C\* than expected e.g. using Li et al. (2016)*
*- Eq. 4-1 gives C\* about as expected by Li et al. (2016)*
*- Compounds used for establishing Eq. 4-2 have HIGHER C\* than expected from the C\* of the compounds used for establishing Eq. 4-1.*

**Response:**

Please refer to our response to Comment #9 from Referee #3.

**Minor specific comments:**

*Abstract:*

*12. I suggest making clearer that grouping into two different O/C regimes was also supported by systematically different thermal desorption behavior (Figs. 3-4).*
*Indeed, I believe this is also a key result that is missing (or unclear) in the abstract.*

**Response:**

We have added this result to the section of Abstract in the revised manuscript, which (Line 23-26) reads, "Among them, 181 organic formulae including 91 CHO and 90 CHON compounds were then selected since their thermograms can be characterized with clear $T_{max}$ values in more than 20 out of 30 filter samples and subsequently divided into two groups according to their O/C ratios and different thermal desorption behavior.".

*Main text:*

*13. General: should briefly go into the difference between a compound's saturation vapor pressure and a*

*compound's effective saturation vapor pressure (or concentration) in regards to partitioning in/out of aerosol particles.*

**Response:**

We now state in the revised manuscript (Line 37-42) that "The effective saturation mass concentration ($C^*$) includes the effect of non-ideal thermodynamic mixing with an activity coefficient ($\gamma$), thus $C^* = \gamma C_0$, and $C^*$ equals $C_0$ under the assumption of ideal thermodynamic mixing (Donahue et al., 2011). Saturation mass concentration is regarded as one of critical physicochemical parameters for organic aerosols' components. The organic compounds with $C^* < 0.1 \ \mu g \cdot m^{-3}$ are mostly in the condensed phase, the organic compounds with $C^* > 1000 \ \mu g \cdot m^{-3}$ are almost entirely in the gas phase, and the organic compounds with $1 < C^* < 100 \ \mu g \cdot m^{-3}$ will be found in both phases under typical conditions (Donahue et al., 2009).".

*14.Lines 56-60 ... If I remember correctly, Tröstl et al. did not know the molecular structures of the observed compositions classified as HOM, nor their saturation concentrations, but guessed the former and correspondingly modeled (using SIMPOL?) the latter. (Subsequently, it also remains unclear here what Stolzenburg et al. were "fitting" to.)*

**Response:**

We now state in the revised manuscript (Line 59-62) that "The molecular structures of these 15 HOMs are unclear, but their saturation concentrations were estimated using the SIMPOL method on the basis of supposed molecular structures (Tröstl et al., 2016). As the covalently bonded dimers are abundant in HOMs from ozonolysis of $\alpha$-pinene, Stolzenburg et al. (2018) fitted parameters using monomer and dimer HOMs separately, allowing a parameter to include the covalent binding.".

*15.Lines 71-74: An important missing piece of information on the FIGAERO procedure is that the desorption temperature is ramped up linearly.*

**Response:**

The missing information have been added to the main text (Line 73-76), which reads, "Basically, the desorption temperature is ramped up linearly, the particulate organic compounds with different vapor pressures are thermo-desorbed and then characterized with distinct thermograms (*i.e.*, desorption signal versus temperature), and the temperature corresponding to the first peak signal ($T_{max}$) correlates with the vaporization enthalpy of a compound (Lopez-Hilfiker et al., 2014).".

*16.Lines 77-80: A weakness of the "second method" is potential measurement artifacts that obscure the true composition of detected species, which thermal desorption methods are prone to. But I would not conclude that the "third method" is generally and necessarily superior, as these lines now seem to suggest.*

**Response:**

Stark et al. (2017) found that compared to the formulae method and partitioning method, the volatility distributions from the thermogram method are likely the closest to the real distributions.

And according to your suggestion, we have revised the wording of this part, which (Line 79-82) reads, "Compared with the parameterization method from organic aerosols' molecular formulae, the thermogram method is able to give a volatility distribution that is likely closer to the real one. Since the molecular formula method likely treats the thermal decomposition products after heating as evaporated

organic molecules, and thus overestimates the overall volatility of a group of organics (Stark et al., 2017).".

17. *Lines 94-95: I believe some text is redundant here (explained twice what volatility is important for).*

**Response:**
We have removed redundant sentences in our manuscript accordingly.

18. *Line 100: I would explicitly mention how (by which of the 3 methods) was "C\* measured".*

**Response:**
This has been explicitly mentioned in main text (Line 102-104), which reads, "In addition, we developed empirical volatility-molecular formula functions based on selected CHO and CHON compounds with varying O/C ratios from ambient OA particles. The *C\** of these selected compounds were estimated by obtained $T_{max}$ from thermograms.".

19. *Line 110: Unclear how thermal desorption was performed. There was a flow of 2.3 lpm and one of 1.0 lpm. If only 1.0 lpm went through the filter and into the IMR, what happened to the remainder 1.3 lpm flow, and what was it for?*

**Response:**
A schematic diagram of desorption gas heater unit of FIGAERO is shown in Figure R12. The 1.0 lpm out of the 2.3 lpm flow was sampled through a dedicated orifice into the IMR. The remainder of 1.3 lpm flow returned through the gap between 1/4 OD tube and 1/2 OD tube and was discharged as shown in Figure R12.

[Figure]

**Figure R12.** A schematic diagram of the FIGAERO desorption gas heater unit ("st.st" is for stainless steel). (Bannan et al., 2019)

20. *I would also clarify in this paragraph how the filter was heated -- or rather that (presumably) it was the UHP N2 that would pass through the filter that was heated. In this regard: where and how was the nominal desorption temperature measured?*

**Response:**
In Figure R12, two 150 W cartridge heaters are used to heat a copper block that is connected with a 1/4 inner OD copper tube. The nitrogen desorption gas is heated when it goes through the hot copper section, and then the particles on the filter are thermally desorbed by the heated nitrogen. The temperature above the filter is measured by the long thermocouple shown here in blue in Figure R12 on the surface of the

Teflon filter (Bannan et al., 2019).

Since the measurement of the desorption temperature has been described in detail by Bannan et al. (2019), we cited the corresponding papers instead of adding the detailed descriptions in our revised manuscript, which (Line 111-112) reads, "The design and operation of the FIGAERO have been introduced in previous studies (Bannan et al., 2019; Lopez-Hilfiker et al., 2014; Thornton et al., 2020; Ye et al., 2020)".

*21.Lines 113-115: Why was 134 °C chosen as the highest temperature? Typically, FIGAERO is operated using desorption temperatures up to 200 °C. A shorter ramp could make sense if going primarily for Tmax, as most "nice" peaks would probably occur before 134 °C. On the other hand, I would be worried about accumulating organic material on the filter, which might cause measurement artifacts...*

**Response:**

Indeed, FIGAERO was operated using desorption temperatures up to 200 °C in previous studies. However, the highest temperature above the filter only reached 134°C in our laboratory experiments, although the maximum temperature of the heating copper block can be set to around 300°C. This could be due to aging of the heating block, and/or insufficient heat transfer between the heater and the nitrogen gas.

The collected ambient samples were analyzed with the FIGAERO offline, and there were different filters for each thermal desorption procedure. Moreover, the filter was held at 134°C for 40 min, which can ensure most of the organic compounds evaporate. As a result, these would not cause organic material to accumulate on the filter and would also not influence measuring the next filter.

We now state in the revised manuscript (Line 122-125) that "Most of the ambient organic compounds can be desorbed from the filter at less than 134°C (Huang et al., 2019). Furthermore, high molecular weight organic compounds (e.g. $C_{27}H_{52}O_4$ ) can be evaporated from the filter below 120 °C (Wang et al., 2016). Therefore, the highest temperature of 134 ˚C is feasible in our study.".

*22.Line 116: How was the "blank filter" used for obtaining backgrounds?*

**Response:**

We have described how to obtain backgrounds using the blank filter in the revised manuscript, which (Line 126-128) reads, "The blank filter was analyzed by the same thermal desorption procedure as that of the field samples. The obtained signals are treated as the background signals.".

*23.Also: What desorption temperature ramp rate was used? Thornton et al. (2020) and Ylisirniö et al. (2021), e.g., have suggested that that ramp rate may affect the thermograms and hence Tmax.*

**Response:**

The ramping rate for heating was 2.27 °C/min.

Please also refer to our response to Comment # 4 from Referee #1.

*24.Section 2.2: Was the desorption procedure (as described in 2.1) for the calibration experiments the same as for the field measurements?*

**Response:**

The desorption procedure for the calibration experiments was the same as that for the field filter samples. We have replenished this information in the revised manuscript (Line 125-126) that "The desorption

procedure for the calibration experiments and the field measurements were the same, as shown in Figure S1.".

Figure S1 has been added to the revised SI (Line 19-20) which is also showed as Figure R13 in the response.

[Figure]

**Figure R13.** The desorption procedure for the calibration experiments and the field measurements.

*25.Fig. 1: I believe the caption could be considerably shortened, using reference to Table 1. Table 1 plus the legend of Fig. 1 contain most of the information given in the caption.*

**Response:**
We have shortened the caption of Figure R2 (i.e., Figure 1 in the revised manuscript).
Please also refer to our response to Comment #7 from Referee #1.

*26.Lines 224-228: I would opt to disagree with the conclusion that the reason that Tmax for calibration set #5 were higher than for set #4 was due to the ammonium sulfate. I would instead rather argue the effect was due to increased filter loading: 1000 ng for set #5 vs. 200 ng for set #4. 1000 ng is clearly in the range of filter loadings that previous studies have seen increased Tmax for that were argued to arise from matrix effects, specifically, I believe, loss of relative surface area available for desorption (Huang et al. 2018; Thorntonet al., 2020).*
*Both effects might play a role, but I do not think they can be separated here.*
*Oh, reading on, I see that this reason is in fact brought up towards the end of the paragraph. I would consider reformulating or restructuring the paragraph to improve clarity.*
*However, I still disagree with the wording in Line 232 ("In other words,..."). Enhanced interactions between aerosol components could lower the effective C\* also without increasing viscosity. Again, reading on, I believe the authors would agree with that, but the way viscosity is brought up may be*

*confusing...*

**Response:**

According to other referees' suggestions, we have conducted more laboratory experiments (No.5-6 and No.8-11 in Table 1 in the revised manuscript) to investigate the influences of ammonium sulfate and mass loadings. We have added more discussion in the revised manuscript.

Please also refer to our response to Comment #8 from Referee #1.

*27.Lines 244-247: I am not sure I am following. Also, judging from Ylisirniö et al. (2021), more information on the calibration procedure used by Nah et al. (2019b) would be needed to ascertain that the cyan dash-dot and solid black lines in Fig. 1 should actually be compared.*

**Response:**

The cyan dash-dot line was obtained by Nah et al. (2019b) with acids and erythritol, and the solid black line was obtained with PEGs in this study. And both these two lines were obtained by the syringe deposition method. There is no more information on the calibration procedure used by Nah et al. (2019b) other than the standards and method. Therefore, we have rewritten this sentence. Please refer to our response to Comment #12 from Referee #2.

*28.Lines 254-261 (last paragraph in 3.1): I think the argument for using calibration #5 for the analyzing the ambient measurements is sound, in principle, and would deliver the most accurate estimates for C\* here.*

*However, it could not be only the deposited mass on the filter that affects Tmax, but also the particle (mass) size distribution of the deposited aerosol. I wonder therefore also how those distributions compared between calibration experiments and ambient samples?*

**Response:**

Thanks for pointing out this issue. Indeed, particle size has a moderate impact on the measured $T_{max}$ values of organic compounds (Ylisirniö et al., 2021). The particle size distributions of calibration experiments and ambient aerosols are similar to each other and have been added in the revised manuscript, as shown in Figure R6 (i.e., Figure S8 in the revised SI).

To clarify this issue, we now state in the revised manuscript (Line 270-272) that "The particle size distributions and peak diameters of polydisperse particles in our laboratory experiments (No.4, No.6 and No.7) are similar to those of the ambient samples (Figure S8). Therefore, in our study, particle size distributions have a minor effect on measured $T_{max}$.".

*29.Lines 274-276: For clarity, I would re-iterate, which peak was chosen for obtaining Tmax in cases where there was more than one fitted peak. I.e., was it always the cooler one, as indicated in Section 2? (Fig. S5d shows an interesting example where of a wide lower cooler peak, and a sharper higher hotter peak. Was it still the cooler one that was chosen, despite being lower?)*

*Oh, peaking forward, I see that Tmax of both or all peaks were actually considered, at least at first. So this comment may be moot. Still, these lines could already be clearer in respect to how (and which) Tmax were obtained and used for further analysis.*

**Response:**

The cooler ones were always chosen for $T_{max}$ in cases where there was more than one fitted peak, even if the signal of the cooler peak was lower than that of the hotter peak.

*30.Line 307 (and in general): It seems to be suggested, as also suggested in previous instances in the text, that the higher Tmax obtained from thermograms exhibiting double peaks have rigorously been attributed to arising thermal decomposition, rendering them unusable for further analysis for assessing C\*. That may be correct reasoning in many instances, but conceivably not always (e.g., for the case of isomers with substantially, but not necessarily unreasonably, different C\*). It may be of interest to at least initially keep those hotter peaks "in the game", and discard them at a later point, as the authors' further analysis would probably be able to make a much stronger argument pro/contra decomposition being involved.*

**Response:**

We have added data points of those hotter peaks (i.e., second peaks of thermograms) for further analysis, as shown in Figure R14, the warmer peaks of double peaks are mainly attributed to thermal decomposition because the C\* of warmer peaks do not decrease with their corresponding molecular weight like cooler peaks. Hence, the $T_{max}$ for the higher-temperature ones in double-peak thermograms of 78 compounds are not taken into account.

[Figure]

**Figure R14.** Saturation mass concentration of CHO and CHON compounds against their molecular weights, as colour-coded by O/C ratios. Note that compounds with an O/C ratio greater than or equal to 1.0 are marked with the same colour. The single peaks and cooler peaks of double-peaks are denoted by circles and warmer peaks of double-peaks are denoted by triangles. Whiskers denote 25[th] and 75[th] percentile values of measured saturation mass concentration from 30 ambient samples, and whiskers are ultimately due to variability in the measured $T_{max}$ of CHO and CHON compounds. Dashed ellipses group compounds on the basis of O/C range.

*31.Fig. 4: I would clarify in the caption: Are the whiskers ultimately due to variability in the measured Tmax (and hence variability in derived C\*)?*

**Response:**

Indeed, the whiskers are ultimately due to variability in the measured $T_{max}$, which has been clarified in the caption of Figure R1 (i.e., Figure 4 in the revised manuscript).

*32.Line 328-329: Please clarify, are those correlations for the two given compounds average correlations (plus standard deviations), while averaging over the individual correlations with each compound in the respective groups? That's how I understood it, but I could see how I could also be misunderstanding...*

**Response:**

The correlation coefficients were obtained by averaging over the individual correlations of $C_6H_{10}O_5$ / $C_{18}H_{34}O_2$ with all compounds in the red/blue dashed ellipse.

To clarify this point, we have described the correlations in more detail in the revised manuscript (Line 348-350), which reads, "The correlation coefficients (*Pearson's r*) between $C_6H_{10}O_5$ and 92% of compounds in the red dashed ellipse are from 0.64 to 0.98, and the correlation coefficients between $C_{18}H_{34}O_2$ and 37% of compounds in the blue dashed ellipse are from 0.60 to 0.74.".

*33.Line 338: Please remind the reader on which volatility range the fits/parametrizations in Mohr et al. are based on?*

**Response:**

We have added the volatility range of compounds used to fit parametrizations by Mohr et al. (2019) to the revised manuscript (Line 358-361), which reads, "Mohr et al. (2019) derived parameterization mainly based on HOMs ($-11 < \log_{10}(C^*) < 3$) produced by α-pinene oxidation, whereas our fits are mainly based on semi-volatility organic compounds (SVOCs, $10^{-0.5} < C^* \le 10^{2.5}$) and low-volatility organic compounds (LVOCs, $10^{-4.5} < C^* \le 10^{-0.5}$), which are predominantly in the particle phase in the atmosphere.".

*34.Table 2: Please include also values that have been used in/suggested by the cited literature!*

**Response:**

We have added the values that have been used and suggested by the cited literature to Table R4 (i.e., Table 2 in the revised manuscript).

**Table R4.** The volatility parameterizations of this study and cited literature. In this study, the parameterizations of saturation mass concentration were modified by the least-square optimization from Eq. (4) at 298 K.

| | $n_c^0$ | $b_c$ | $b_o$ | $b_{co}$ | $b_N$ | Suggested O/C range | $b_{add}$ |
|---|---|---|---|---|---|---|---|
| Eq. (4-1) in this study | 25 | 0.0700 | 0.6307 | -0.0615 | 2.3962 | 0.25-1 | / |
| Eq. (4-2) in this study | 25 | 0.2075 | 2.8276 | -1.0744 | 1.8223 | 0-0.25 | / |
| Donahue et al. (2011) | 25 | 0.475 | 2.3 | -0.3 | / | / | / |
| Mohr et al. (2019) | 25 | 0.475 | 0.2 | 0.9 | 2.5 | / | / |
| Stolzenburg et al. (2018) (monomers) | 25 | 0.475 | 2.3 | -0.3 | / | / | 0.90 |
| Stolzenburg et al. (2018) (dimers) | 25 | 0.475 | 2.3 | -0.3 | / | / | 1.13 |

| | | | | | | | |
|---|---|---|---|---|---|---|---|
| Li et al. (2016) (CHO) | 22.66 | 0.4481 | 1.656 | -0.7790 | / | / | / |
| Li et al. (2016) (CHON) | 24.13 | 0.3667 | 0.7732 | -0.07790 | 1.114 | / | / |

*35.Line 353: Was there any specific kind of OOA that Donahue et al. (2011) referred to corresponding to the yellow-dashed box in Fig. S7?*

**Response:**

Donahue et al. (2011) did not refer any specific kind of OOA to correspond with the yellow-dashed box in Figure. S11. Note that "Figure S7" has been changed to "Figure S11" in the revised manuscript.

*36.Line 362: "accuracy" in which respect? Please clarify.*
**Response:**
The "accuracy" is in respect of the predication of volatility.

To clarify this word, we now state in the revised manuscript (Line 384-385) that "On the other hand, the accuracy in predication of volatility of the parameterizations of Stolzenburg et al. (2018), Li et al. (2016), and Eq. (4-1) is generally comparable (Figure 5a).".

*37.Line 366: What is "therefore" referring to?*

**Response:**

Since we have added more CHO compounds (O/C: 0.25-1) to compare the parameterizations with the SIMPOL method, the detailed description of Figure 5a has been rewritten.

Please also refer to our response to Comment #6 from Referee #3.

**Referee #4**

*The study conducted by Ren et al. is looking into FIGAERO-I-CIMS thermograms, more precisely the maximum temperatures of the first peaks of the thermograms (Tmax). The study starts with a suite of laboratory experiments with PEG samples of different volatilities which they either inject on the FIGAERO filter using a syringe or nebulize, dry, dilute and collect onto the FIGAERO filter. They attain similar results as Ylisirniö et al. (2021), reproducing a quantitatively similar relationship between saturation vapor pressure (Psat) and Tmax. Ren et al. investigate this relationship further by making mixtures of various PEG, citric acid and erythritol with ammonium sulfate (AS), either as a mixture between one organic component and AS or as one mixture between all organic components and AS. These were then deposited on the FIGAERO filter following the atomizer method. A discrepancy could be observed between the Psat -Tmax relationships derived from these two experiment types. They attribute the mismatch to either AS derived effects such as organic salt formation, viscosity limitations in evaporation or matrix effects. Finally, the authors utilize the Psat -Tmax relationship derived from the mixture calibration involving all organic compounds to calculate saturation vapor concentrations (C\*) for their field data. They further noticed that these Tmax derived C\* displayed as a function of molecular weight showed two groups/clusters with characterized with different O:C-ratios. The authors finally derive two molecular formulae based C\* parameterizations for these groups, respectively. The results are compared to other molecular formulae-based C\* parameterizations.*

*The manuscript could be potentially very useful for the FIGAERO community if the calibration results were investigated further and the reasons behind the mismatch could be narrowed down and the potential influence of inorganic salts or matrix effects on Tmax and therefore C\* in the field could be assessed. It would be useful if the authors could carefully evaluate how reliable the C\* derived from Tmax are under environments with high mass loading and inorganic salt concentrations. I find this necessary before parameterizations are being derived from these Tmax – C\* relationships. I recommend publication after major revisions.*

**Response:**
We are very grateful for the insightful comments from Reviewer #4 and have revised our manuscript accordingly.

**Main comments:**

*1.I think it would be crucial to understand/narrow down what actually caused the change in the Psat - Tmax relationship when comparing the line derived from the single PEG+AS mixtures vs one solution. I would suggest you to perform more laboratory experiments such as: 1. A single organic mixture (all organic compounds included) without AS, 1000 µg deposited on filter with the atomizer method; 2. Replicating No.5 experiments and probing the effect of mass loading.*

**Response:**
According to your suggestions, we have performed more laboratory experiments (No.6 and No.8-11 sets of experiments in the revised manuscript) to investigate the effects from AS, mass loadings and matrix

effects within organics. Besides, we also added No.5 (original No.5 is set as No.7) set of experiments (i.e., atomizing 500 ng AS + 500 ng Organics (each)) suggested by Reviewer #1.

Please refer to our response to Comment #8 from Referee #1 for detailed information.

*2.The authors should also provide information of the size distribution and temperature for the different experiments, especially for No.4 and No.5. It should be noted that Ylisirniö et al. (2021) mention their significant role in causing discrepancies in the Psat -Tmax relationship.*

**Response:**

We have added information on the size distributions to the revised manuscript. Please refer to our response to Comment #28 from Referee #3.

Furthermore, the ramping rate for heating (temperature) was 2.27 °C/min for calibration experiments and field measurements, as shown Figure R13.

Please also refer to our response to Comment #4 from Referee #1.

*3.The authors should at least provide the 95% credible intervals along the fitted lines in Figure 1 if showing all the data points (replicates) decreases the readability of the graph. It would be useful to see how much scatter there is between replicates. This scatter could even hold some information about possible loading effects.*

**Response:**

We have provided the 95% credible intervals along with the fitted lines in Figure R5 (i.e., Figure S7 in the revised SI). We did not add the 95% credible intervals directly to Figure 1 because this would decrease the readability of the graph.

We now state in our revised manuscript that (Line 260-261) "The 95% credible intervals of No.5, No.6 and No.7 experiments are significantly larger than the others, which may be attributed to their higher mass loading (1000 ng) than those in other experiments (100 ng, 200 ng and 500 ng) (Figure S7).".

*4.The authors should think about providing a schematic of their laboratory setup. They mention the possibility of organic salt formation in No.5 mixture that could cause the increase in the observed Tmax values when compared to No.4 experiments. If the sample is dried immediately after the nebulizer there is not much time for any organic salt formation under favorable conditions (high humidity and aerosol liquid water content). Do the authors think there is time for such reactions to actually happen?*

**Response:**

The schematic of laboratory setup has been provided in the Figure R15 (i.e., Figure S3 in the revised SI). We agree with the reviewer that there is not much time for organic salt formation in our calibration experiments. Therefore, the sentence has been modified.

Also, we have rewritten this part based on the results of added calibration experiments. Please refer to our response to Comment #8 from Referee #1.

[Figure]

**Figure R15.** Schematics of the atomization method setup.

*5.After understanding the significance of the matrix effects on the Psat -Tmax relationship in the calibration data – do the authors still recommend deriving Psat from Tmax? Do the authors observe variability in Tmax in the ambient samples that covary with mass loading? How much variability was there in the mass deposited on the 30 filters analyzed and how does it compare to the 1000 μg calibration reference? How would the matrix effects from ambient samples affect the predicted C\*?*

**Response:**
According to the results of laboratory experiments, we found that the matrix effects have an influence on the Psat -$T_{max}$ relationship. Thus, we used the calibration curve from No.7 set of experiments that took the matrix effects into account, to analyze the filed samples. This method could accurately represent the volatility of compounds in ambient particles.
We did not observe obvious variability in $T_{max}$ of some dominated compounds in most our ambient samples under various mass loadings. The mass loadings of the 30 filters in our study are from 200 ng to 3500 ng with a median of 1100 ng which is close to 1000 ng.

Please refer to our response to Comment #1 from Referee #2 for detailed information of these effects on $T_{max}$.

*6.It is unclear to me whether the FIGAERO was measuring in real time during the field campaign or whether the filters collected (described in Sect. 2.3) were measured with the FIGAERO offline. Could you please clarify.*

**Response:**
The collected filters were measured with the FIGAERO offline.
We now state in our revised manuscript (Line 190-191) that "…30 filter samples between January 15, 2019 and January 22, 2019 were analyzed with FIGAERO offline, …".

**Minor/technical comments:**

*1.L34: is the Nizkorodov et al. (2011) the best reference for this statement?*

**Response:**

We have replaced "(Nizkorodov et al., 2011)" with "(Jimenez et al., 2009)".

*2.L40-: The descriptions of the past methodologies are incomplete. The description is for example missing classic work with thermodenuders (TD) without CIMS that have been mounted as part of tandem volatility differential mobility analyzers (TDMA) or coupled with an AMS (TD-AMS). The introduction also lacks description of the way the thermograms measured by the TDMA or TD-AMS are being modelled to gain information of C\* or VBS (see for example Cappa, 2010, in Atmos. Meas. Tech or Cappa and Jimenez, 2010, in Atmos. Chem. Phys.). In addition, dilution experiments among many others should not be forgotten. The authors seem to be citing more the work for predicting equilibrium partitioning coefficients than actual C\* measurements. If the authors wish the provide a list of methods used previously, they should cite more relevant literature and make sure to include a complete description. Alternatively, they could focus on describing just the methods relevant for CIMS.*

**Response:**

According to your suggestions, we tend to focus on describing just the methods relevant to CIMS.

We now state in our revised manuscript (Line 44-46) that "During the past years, two major methods relevant to the Chemical Ionization Mass Spectrometer (CIMS) have been developed to characterize the volatility of aerosols. The first one estimates the volatility of an organic species based on its molecular formula.",

and (Line 66) that "The second one estimates the volatility of an organic species on the basis of its desorption thermogram.".

*3.L322-L323: Figure 4 does not contain a red or blue dashed circles, maybe ellipse would be a better word. I found this confusing at first.*

**Response:**

We have replaced "circle" with "ellipse".

[revised manuscript text omitted]

---

## Author Response (AR2)

**A point-to-point response to reviewers' comments**

We are very grateful for the helpful and insightful comments from the reviewers, and have carefully revised our manuscript accordingly. In the following point-to-point response, reviewers' comments are repeated in ***italics***, whereas our responses are in plain texts labelled with **[Response]**. Line numbers in the responses correspond to those in the revised manuscript (the version with all changes accepted). Modifications to the manuscript are in blue.

**Reviewer #4**

*The authors have addressed well the points brought up by the four reviewers. The manuscript has improved significantly and I am happy with the majority of the responses. However, as the authors now report the slow temperature ramping rate in their FIGAERO setup, I realized that the manuscript lacks some discussion regarding the potential effects of thermal decomposition on their observations.*

**Response:**

We are very grateful for the insightful comments from Reviewer #4 and have revised our manuscript accordingly.

*1.It should be noted that a single-peak thermogram does not always ensure that decomposition did not take place. For example, in Riva et al. (2019), SOA (alpha-pinene ozonolysis) formed onto acidic sulfate particles (a chamber study) sampled with a FIGAERO-I-CIMS had a rather unimodal thermogram with a low Tmax, but isothermal evaporation studies on similar SOA formation showed that the SOA was actually of very low volatility. The SOA measured by the FIGAERO during the Riva et al. (2019) experiments had high contributions from high double bond equivalent (DBE) compounds. Discussion about high DBE and thermal decomposition likelihoods were brought up also in Yang et al. (2021). How would Figures 3 and 4 look like if color-coded by DBE? Could you add some discussion about the potential effects of thermal decomposition and how it would affect your volatility parameterizations? Is it possible that the second ellipse (the one at generally higher molecular weights with lower O:C) in Figures 3 and 4 is greatly affected by thermal decomposition and the Tmax is significantly reduced similarly as happened in the Riva et al. (2019) experiments mentioned above?*

**Response:**

Indeed, according to Riva et al. (2019), thermal decomposition of the oligomers in organic aerosols can lead to a misinterpretation of the SOA volatility. Riva et al. (2019) observed an increase in high DBE compounds, which were postulated to be thermal decomposition fragments of high-molecular-weight organic molecules. Besides, those thermal decomposition fragments with high DBEs could lead to an overestimation of

organic aerosol volatility. In our study, the components of ambient organic aerosols are very complex, and thus it is difficult to identify the decomposition products and their parental organic molecules based on thermograms and to quantify this effect on our volatility parameterizations.

Furthermore, Yang et al. (2021) found that the nO, DBE, and $T_{max}$ of the parental compounds may be related to the degree of thermal decomposition by observing 29 standard compounds that include alcohols, monoacids, diacids, polyacids, and multifunctional acids. When DBE $\geq 2$, $T_{max} \geq 72$ °C and nO $>4$, thermal decomposition of parental compounds can occur considerably. Since the parental compounds in ambient organic aerosols cannot be identified in this study, these criteria cannot be directly adopted to predict the degree of thermal decomposition of complex ambient organic aerosols.

Figure 3 and Figure 4 are now colour-coded by DBE, as shown in Figure R1 and Figure R2 (i.e., Figure S11 in the revised Supporting Information (SI)). In Figure R1 and Figure R2, the DBE distribution of these CHO and CHON compounds is random, suggesting that the effect of thermal decomposition on our volatility parameterizations may be minor.

We have added this point to the revised manuscript, which (Line 327-332) reads, "Furthermore, thermal decomposition of the oligomers in organic aerosols can lead to a misinterpretation of the SOA volatility, and double bond equivalent (DBE) has been used to determine the thermal decomposition degree of an individual compound and SOAs formed from the ozonolysis of α-pinene (Riva et al., 2019; Yang et al., 2021). The contents in Figure 4 are further colour-coded by DBE, instead of O:C, as shown in Figure S11. In Figure S11, the DBE distribution of these 181 compounds is random, thus the thermal decomposition could have a minor effect on $T_{max}$.".

The second ellipse (the one at generally higher molecular weights with lower O:C) covers 47 compounds. As shown in Figure R1 and Figure R2, the DBE of most compounds (32 out of 47) in the second ellipse is smaller than or equal to 2, and there are 15 of 47 compounds with high DBE (DBE $>2$). Moreover, the DBE distribution of compounds in the second ellipse has no regularity. Therefore, the $T_{max}$ values of compounds in the second ellipse are less affected by thermal decomposition.

[Figure]

Figure R1. Evaporation and decomposition of 91 CHO and 90 CHON compounds, as colour-coded by DBE. Note that compounds with DBE equal to or greater than 7 are marked with the same colour. The signal peaks, the cooler and warmer peaks of double-peaks are denoted by circles, triangles and diamonds, respectively. Rectangular bands depict the temperature zones in which peaks appear.

[Figure]

Figure R2. Saturation mass concentration of CHO and CHON compounds against their molecular weights, as colour-coded by DBE. Note that compounds with DBE equal to or greater than 7 are marked with the same colour. The CHO and CHON compounds are denoted by squares and hexagrams, respectively. Whiskers denote 25th and 75th percentile values of measured saturation mass concentration from 30 ambient samples and whiskers are ultimately due to variability in the measured $T_{max}$ of CHO and CHON compounds.

*2.You mention on L136 that a slower heating rate leads to a smaller number of decomposition products. Based on Yang et al. (2021), shouldn't it say the opposite or am I missing something?*

**Response:**

Indeed, a slower heating rate leads to a larger decomposition degree of parental compounds. But Yang et al. (2021) also found that for the citric acid, there is a smaller number of products formed after the desorption process under a slower heating rate (i.e., only the primary and secondary dehydration products with no decarboxylation product).

To clarify this point, we now state in our revised manuscript (Line 124-127) that "Also, a slower ramping rate can separate compounds with similar volatilities better (Lopez-Hilfiker et al., 2014). Note that the decomposition degree of parental compounds under a slower ramping rate is higher than that under a faster ramping rate, but a slower heating rate generally leads to a smaller number of thermal decomposition products (Yang et al., 2021).".

**Reviewer #3**

*I would like to thank the authors for carefully considering and addressing my comments (and those of my co-reviewers).*
*I believe the manuscript is much improved and clearer. As in my first review, I still think the manuscript overall is a carefully worked out and useful piece in our quest towards figuring out the volatility of SOA, so I recommend to accept it for publication. However, I also recommend the following minor and technical revisions:*

**Response:**
We appreciate the insightful and positive comments from Reviewer #3 and have revised our manuscript accordingly.

**Minor comments:**

*1.Lines 379+:*
*If I understand correctly, neither Zhao et al. (2013) nor Mazzoleni et al. (2010) report molecular structures, "just" molecular formulae. Please explain or correct.*
*However, some assumptions on molecular structures are indeed necessary to obtain vapor pressures from SIMPOL, so I wonder which assumptions were used here to arrive at the SIMPOL predictions (as in Fig. 5). I may just be missing a step here, which could be useful to point out in the text here. However, that step may be important for interpreting Fig. 5. (Specifically, for 230 of the 245 x-axis values!)*
*Same comment for lines 415+ (dealing with CHON instead of CHO).*

**Response:**
Indeed, neither Zhao et al. (2013) nor Mazzoleni et al. (2010) reported molecular structures. Only molecular formulae were presented in the two studies.
We assume that the molecular structures of compounds from Zhao et al. (2013) and Mazzoleni et al. (2010) are common ones (i.e., most of function groups of their structures are included in SIMPOL, so that their volatilities can be calculated using SIMPOL).

To clarify this issue, we now state in the revised manuscript (Line 384-387) that "The molecular formulae of 230 CHO (O/C: 0.25-1) compounds are from Zhao et al. (2013) and Mazzoleni et al. (2010), and the molecular structures of these 230 compounds are predicted to be common, i.e., most of function groups of their structures are included in SIMPOL, so that the volatilities of these compounds can be estimated by SIMPOL.", and (Line 426-427) that "The molecular structures of selected species are assumed to be common. Then their saturation mass concentration ($C^*$) are estimated by different parameterizations and SIMPOL, respectively.".

*2.Line 383:*
*Who or which parametrization have not been "considering covalent binding"? I suspect*

*that the term "covalent binding" is a too general term here for what the authors try to say. Please double-check and attempt to clarify.*

**Response:**
Mohr et al. (2019) have not considered covalent binding. To clarify this point, we now state in the revised manuscript (Line 388-392) that "Although Mohr et al. (2019) and Stolzenburg et al. (2018) both updated parametrizations based on those 15 HOMs detected by Tröstl et al. (2016), compared to the Mohr et al. (2019) parameterization, the volatility predicted by the Stolzenburg et al. (2018) parameterization does match those by SIMPOL better, which could be attributed to the fact that Mohr et al. (2019) did not separately use parameters for dimer and monomer as Stolzenburg et al. (2018) did, so that the effect of the covalent binding is ignored.".

*3.Line 388:*
*Which "inherent strength and deficiency of the FIGAERO method" is (are) referred to here?*

**Response:**
We now state in our revised manuscript (Line 396-398) that "…, which reflects the inherent strength and deficiency of the FIGAERO method, i.e., the FIGAERO method replies on authentic standards that are commonly LVOCs and is thus less suitable for the more volatile compounds.".

**Technical comments:**

*4.Figure 1:*
*The authors could point out which vapor pressures correspond to which compound, at least for the compounds used in this study. That would be particularly useful because not only PEG-n are used, but also erythritol and citric acid. Also, a slightly discrepant behavior is referred to in the text (line 253), which will be much easier to make out then. ... Oh, that has been done actually in Figs. S4-S6! That helps. It might still help to add some labels like that to Fig. 1 too. I'll leave that to the authors.*
*The other thing, however, is that it is unclear from Fig. 1, which line the "fitted parameters" (legend) correspond to. Please clarify, at least in the caption.*

**Response:**

We have added more labels to Figure R3 (i.e., Figure 1 in the revised manuscript) to point out vapor pressures of corresponding compounds.
We have clarified the "fitted parameters" in the caption of Figure 1, which (Line 233) reads "The fitted parameters correspond to the dark green line.".

[Figure]

Figure R3. Comparison of calibration results obtained in this study with those reported previously. These solid lines denote the calibration results obtained in this study. Error bars represent ± one standard deviation of $T_{max}$ from four replicate experiments. The fitted parameters correspond to the dark green line.

*5.Lines 376:*
*I believe those 15 HOM have not yet been brought up, so I think "the" should be removed.*
*In fact, it should be moved to line 378.*

**Response:**
We have revised our manuscript accordingly.

*6.Line 440:*
*"owning" -> "owing" (I believe)*

**Response:**
We have replaced "owning" with "owing" in the revised manuscript.

*7.Figure S11:*
*I suggest adding a sentence to the caption, explaining which are the compounds explicitly pointed to in the figure. That info is found in the main text, but it would be helpful for the reader to receive some explanation in the figure caption as well.*

**Response:**

We have added a sentence to the caption of Figure S12 in the revised SI, which (Line 59-61) reads "Marked compounds are oleic acid ($C_{18}H_{34}O_2$), levoglucosan or related isomers ($C_6H_{10}O_5$), margaric acid ($C_{17}H_{34}O_2$), linoleic acid ($C_{18}H_{32}O_2$), and palmitic acid ($C_{16}H_{32}O_2$), respectively.".

*8.Figures S13:*
*Please explicitly name the 5 compounds.*
*(Also, "SIMPOL" is missing its I in both axis label and caption.)*

**Response:**
We have explicitly named the 5 compounds in the caption of Figure R4 (i.e., Figure S14 in the revised SI).
And the typos have been revised.

[Figure]

**Figure R4.** Saturation mass concentration (C*) of five organic compounds (i.e., erythritol, PEG-6, PEG-7, PEG-8, and citric acid) from Krieger et al. (2018), Emanuelsson et al. (2016) and Ye et al. (2019) against those calculated by SIMPOL.

**References**

Emanuelsson, E. U., Tschiskale, M. and Bilde, M.: Phase State and Saturation Vapor Pressure of Submicron Particles of meso-Erythritol at Ambient Conditions, J. Phys. Chem. A, 120(36), 7183–7191, doi:10.1021/acs.jpca.6b04349, 2016.

Krieger, U. K., Siegrist, F., Marcolli, C., Emanuelsson, E. U., Gøbel, F. M., Bilde, M., Marsh, A., Reid, J. P., Huisman, A. J., Riipinen, I., Hyttinen, N., Myllys, N., Kurtén, T., Bannan, T., Percival, C. J. and Topping, D.: A reference data set for validating vapor pressure measurement techniques : homologous series of polyethylene glycols, , 49–63, 2018.

Mazzoleni, L. R., Ehrmann, B. M., Shen, X., Marshall, A. G. and Collett, J. L.: Water-Soluble Atmospheric Organic Matter in Fog: Exact Masses and Chemical Formula Identification by Ultrahigh-Resolution Fourier Transform Ion Cyclotron Resonance Mass Spectrometry, Environ. Sci. Technol., 44(10), 3690–3697, doi:10.1021/es903409k, 2010.

Riva, M., Heikkinen, L., Bell, D. M., Peräkylä, O., Zha, Q., Schallhart, S., Rissanen, M. P., Imre, D., Petäjä, T., Thornton, J. A., Zelenyuk, A. and Ehn, M.: Chemical transformations in monoterpene-derived organic aerosol enhanced by inorganic composition, npj Clim. Atmos. Sci., 2(1), 1–9, doi:10.1038/s41612-018-0058-0, 2019.

Stolzenburg, D., Fischer, L., Vogel, A. L., Heinritzi, M., Schervish, M., Simon, M., Wagner, A. C., Dada, L., Ahonen, L. R., Amorim, A., Baccarini, A., Bauer, P. S., Baumgartner, B., Bergen, A., Bianchi, F., Breitenlechner, M., Brilke, S., Mazon, S. B., Chen, D., Dias, A., Draper, D. C., Duplissy, J., Haddad, I. El, Finkenzeller, H., Frege, C., Fuchs, C., Garmash, O., Gordon, H., He, X., Helm, J., Hofbauer, V., Hoyle, C. R., Kim, C., Kirkby, J., Kontkanen, J., Kürten, A., Lampilahti, J., Lawler, M., Lehtipalo, K., Leiminger, M., Mai, H., Mathot, S., Mentler, B., Molteni, U., Nie, W., Nieminen, T., Nowak, J. B., Ojdanic, A., Onnela, A., Passananti, M., Petäjä, T., Quéléver, L. L. J., Rissanen, M. P., Sarnela, N., Schallhart, S., Tauber, C., Tomé, A., Wagner, R., Wang, M., Weitz, L., Wimmer, D., Xiao, M., Yan, C., Ye, P., Zha, Q., Baltensperger, U., Curtius, J., Dommen, J., Flagan, R. C., Kulmala, M., Smith, J. N., Worsnop, D. R., Hansel, A., Donahue, N. M. and Winkler, P. M.: Rapid growth of organic aerosol nanoparticles over a wide tropospheric temperature range, Proc. Natl. Acad. Sci. U. S. A., 115(37), 9122–9127, doi:10.1073/pnas.1807604115, 2018.

Yang, L. H., Takeuchi, M., Chen, Y. and Ng, N. L.: Characterization of thermal decomposition of oxygenated organic compounds in FIGAERO-CIMS, Aerosol Sci. Technol., 55(12), 1321–1342, doi:10.1080/02786826.2021.1945529, 2021.

Ye, Q., Wang, M., Hofbauer, V., Stolzenburg, D., Chen, D., Schervish, M., Vogel, A., Mauldin, R. L., Baalbaki, R., Brilke, S., Dada, L., Dias, A., Duplissy, J., El Haddad, I., Finkenzeller, H., Fischer, L., He, X., Kim, C., Kürten, A., Lamkaddam, H., Lee, C. P., Lehtipalo, K., Leiminger, M., Manninen, H. E., Marten, R., Mentler, B., Partoll, E., Petäjä, T., Rissanen, M., Schobesberger, S., Schuchmann, S., Simon, M., Tham, Y. J., Vazquez-Pufleau, M., Wagner, A. C., Wang, Y., Wu, Y., Xiao, M., Baltensperger, U., Curtius, J., Flagan, R., Kirkby, J., Kulmala, M., Volkamer, R., Winkler, P. M., Worsnop, D. and Donahue, N. M.: Molecular Composition and Volatility of Nucleated Particles

from α-Pinene Oxidation between -50 °c and +25 °c, Environ. Sci. Technol., 53(21), 12357–12365, doi:10.1021/acs.est.9b03265, 2019.

Zhao, Y., Hallar, A. G. and Mazzoleni, L. R.: Atmospheric organic matter in clouds: Exact masses and molecular formula identification using ultrahigh-resolution FT-ICR mass spectrometry, Atmos. Chem. Phys., 13(24), 12343–12362, doi:10.5194/acp-13-12343-2013, 2013.

---

## Author Response (AR3)

**A point-to-point response to editor's comments**

We are very grateful for the helpful and insightful comments from the editor, and have carefully revised our manuscript accordingly. In the following point-to-point response, editor's comments are repeated in *italics*, whereas our responses are in plain texts labelled with **[Response]**. Line numbers in the responses correspond to those in the revised manuscript (the version with all changes accepted). Modifications to the manuscript are in blue.

*1. Yang et al. illustrated how the screening criteria combined with thermogram data can be used to examine the presence of thermal decomposition in ambient organic aerosol datasets (Figure 8 in Yang et al. and associated discussions). Could you make similar evaluations to further support the statement that the effect of thermal decomposition is minor?*

**Response:**
Yang et al. (2021) identified in their ambient dataset that $C_5H_8O_6$ met the screening criteria for considerable thermal decomposition (DBE $\geq$2, $T_{max}$ $\geq$72 °C and nO >4) by investigating thermograms of the parental compound ($C_5H_8O_6$) and possibly its decarboxylation product ($C_4H_8O_4$) to confirm the presence of thermal decomposition.

In our case, we analyzed 181 compounds (91 CHO and 90 CHON) according to the screening criteria of Yang et al. (2021). Thermograms of 78 compounds show bimodal peaks, most of the warmer of which correspond to thermal decomposition of higher molecular weight organic compounds (Huang et al., 2018) but not included in the following analysis.

Therefore, we used the screening criteria of Yang et al. (2021) to screen the remaining 103 compounds with unimodal thermograms. Among the 103, there are 33 compounds that meet the screening criteria for considerable thermal decomposition (DBE $\geq$2, $T_{max}$ $\geq$72 °C and nO >4), and the thermograms of these 33 compounds generally present a single-peak without broad tailing and fronting, as shown in Figure R1a and c (i.e., Figure S9a and c in the revised Supporting Information).

We then investigated possible thermal decomposition products of these 33 compounds. Although thermal decomposition could be very complex, here we only considered dehydration products and decarboxylation products. Two compounds were identified to be potential thermal decomposition products, i.e., $C_9H_{17}O_3N$ and $C_{10}H_{13}O_4N$ could be a decarboxylation product of $C_{10}H_{17}O_5N$ and a secondary dehydration product of $C_{10}H_{17}O_6N$, respectively. $T_{max}$ of $C_9H_{17}O_3N$ was higher than that of $C_{10}H_{17}O_5N$ by about 8 °C and $T_{max}$ of $C_{10}H_{13}O_4N$ was higher than that of $C_{10}H_{17}O_6N$ by about 4 °C, which is close to the results of Yang et al. (2021). However, this observation can also be explained by isomers with vastly different vapor pressures (Huang et al., 2018; Lopez-Hilfiker et al., 2015). For a quick identification of these compounds, $C_9H_{17}O_3N$,

$C_{10}H_{17}O_5N$, $C_{10}H_{13}O_4N$ and $C_{10}H_{17}O_6N$ have been marked in Figure R2 (i.e., Figure 3 in the revised manuscript).

The O/C ratios of $C_9H_{17}O_3N$ and $C_{10}H_{13}O_4N$ are in the range of 0.25-1, we fitted the volatility parameterization, and obtained Eq. (4-R3) using compounds in the left dashed ellipse in Figure R2 that do not contain $C_9H_{17}O_3N$ and $C_{10}H_{13}O_4N$, and Eq. (4-1) by compounds in the left dashed ellipse in Figure R2 that contain $C_9H_{17}O_3N$ and $C_{10}H_{13}O_4N$. We used measured saturation mass concentration as benchmarks to compare the quality of Eq. (4-1) and Eq. (4-R3). It turned out that the volatilities of compounds in the left dashed ellipse in Figure R2 estimated by Eq. (4-1) are very close to those by Eq. (4-R3), as shown in Figure R4. Thus, the effect of potential thermal decomposition of our ambient sample is minor.

We have added this point to the revised manuscript, which (Line 332-342) reads, "On the other hand, 33 out of the 103 unimodal compounds meet the screening criteria of Yang et al. (2021) for considerable thermal decomposition (DBE $\geq 2$, $T_{max} \geq 72$ °C and nO >4), and the unimodal thermograms of these 33 compounds generally do not present broad tailing and fronting, as shown in Figure S9a and c. The possible thermal decomposition products of these 33 compounds are then investigated. Although thermal decomposition could be very complex, here we only considered dehydration products and decarboxylation products. $C_9H_{17}O_3N$ and $C_{10}H_{13}O_4N$ could be a decarboxylation product of $C_{10}H_{17}O_5N$ and a secondary dehydration product of $C_{10}H_{17}O_6N$, respectively. $T_{max}$ of $C_9H_{17}O_3N$ is higher than that of $C_{10}H_{17}O_5N$ by about 8 °C and $T_{max}$ of $C_{10}H_{13}O_4N$ is higher than that of $C_{10}H_{17}O_6N$ by about 4 °C, which is close to the results of Yang et al. (2021). However, this observation can also be explained by isomers with vastly different vapor pressures (Huang et al., 2018; Lopez-Hilfiker et al., 2015). For a quick identification of these compounds, $C_9H_{17}O_3N$, $C_{10}H_{17}O_5N$, $C_{10}H_{13}O_4N$ and $C_{10}H_{17}O_6N$ have been marked in Figure 3. Since only two compounds may be thermal decomposition fragments, thermal decomposition likely has little effects on our subsequent analysis.".

[Figure]

**Figure R1.** Four representative types of thermograms. Red, green and blue lines represent fitting curves for the overall thermogram, the first desorption peak, and the second desorption peak, respectively.

[Figure]

**Figure R2.** Evaporation and decomposition of 91 CHO and 90 CHON compounds, of which the reagent ion (I⁻) is excluded from their formulae. The signal peaks, the cooler and warmer peaks of double-peaks are denoted by red, green and purple circles, respectively. Rectangular bands depict the temperature zones in which peaks appear.

[Figure]

**Figure R3.** Saturation mass concentration ($C^*$) of compounds in the left dashed ellipse in Figure R2 as estimated by the Eq. (4-1) and Eq. (4-R3) against.

*2. Suggest to remove the word generally in "... but a slower heating rate generally leads to a smaller number of thermal decomposition products (Yang et al., 2021)." since only citric acid was tested.*

**Response:**
The word "generally" has been removed in the revised manuscript.

**References**

Huang, W., Saathoff, H., Pajunoja, A., Shen, X., Naumann, K. H., Wagner, R., Virtanen, A., Leisner, T. and Mohr, C.: α-Pinene secondary organic aerosol at low temperature: Chemical composition and implications for particle viscosity, Atmos. Chem. Phys., 18(4), 2883–2898, doi:10.5194/acp-18-2883-2018, 2018.

Lopez-Hilfiker, F. D., Mohr, C., Ehn, M., Rubach, F., Kleist, E., Wildt, J., Mentel, T. F., Carrasquillo, A. J., Daumit, K. E., Hunter, J. F., Kroll, J. H., Worsnop, D. R. and Thornton, J. A.: Phase partitioning and volatility of secondary organic aerosol components formed from α-pinene ozonolysis and OH oxidation: The importance of accretion products and other low volatility compounds, Atmos. Chem. Phys., 15(14), 7765–7776, doi:10.5194/acp-15-7765-2015, 2015.

Yang, L. H., Takeuchi, M., Chen, Y. and Ng, N. L.: Characterization of thermal decomposition of oxygenated organic compounds in FIGAERO-CIMS, Aerosol Sci. Technol., 55(12), 1321–1342, doi:10.1080/02786826.2021.1945529, 2021.